# Genomic profiling and pre-clinical modelling of breast cancer leptomeningeal metastasis reveals acquisition of a lobular-like phenotype

Amanda Fitzpatrick [1,2], Marjan Iravani [1], Adam Mills [1], David Vicente [1], Thanussuyah Alaguthurai[3], Ioannis Roxanis[1], Nicholas C. Turner [1,4], Syed Haider [1], Andrew N. J. Tutt [1,3,5] & Clare M. Isacke [1] ✉

Breast cancer leptomeningeal metastasis (BCLM), where tumour cells grow along the lining of the brain and spinal cord, is a devastating development for patients. Investigating this metastatic site is hampered by difficulty in accessing tumour material. Here, we utilise cerebrospinal fluid (CSF) cell-free DNA (cfDNA) and CSF disseminated tumour cells (DTCs) to explore the clonal evolution of BCLM and heterogeneity between leptomeningeal and extra-cranial metastatic sites. Somatic alterations with potential therapeutic actionability were detected in 81% (17/21) of BCLM cases, with 19% detectable in CSF cfDNA only. BCLM was enriched in genomic aberrations in adherens junction and cytoskeletal genes, revealing a lobular-like breast cancer phenotype. CSF DTCs were cultured in 3D to establish BCLM patient-derived organoids, and used for the successful generation of BCLM in vivo models. These data reveal that BCLM possess a unique genomic aberration profile and highlight potential cellular dependencies in this hard-to-treat form of metastatic disease.

Despite advances in effective treatment of visceral metastatic disease, central nervous system (CNS) metastasis increasingly limits patient survival[1,2]. Leptomeningeal metastasis, where tumour cells spread over the meninges lining the brain and spinal cord and shed into the surrounding cerebrospinal fluid (CSF), most commonly occurs in breast cancer, lung cancer and melanoma[3,4]. Leptomeningeal metastasis causes debilitating neurological symptoms with often rapid clinical deterioration. Median survival in breast cancer leptomeningeal metastasis (BCLM) is 3.4–5.4 months despite current treatments[4–6]. Published BCLM cohorts show an enrichment for invasive lobular cancer (ILC)[7–9], the most common 'special' subtype of breast cancer[10,11]

which is characterised histologically by discohesive tumour cells arranged in single files or as individual cells, driven predominantly by mutational inactivation of *CDH1* (E-cadherin)[12,13]. By contrast, the more common invasive carcinoma of no special type (IC-NST) are E-cadherin proficient and exist in nests or sheets of cells with preserved cell-cell adhesions[14].

Although there is now substantial knowledge of the genomic events that evolve in metastatic breast and other solid cancers[15–20], the biology of leptomeningeal metastasis remains poorly understood since the leptomeninges are a rarely-biopsied site. It is often cited that BCLM is a late manifestation of widespread metastatic disease,

[1]The Breast Cancer Now Toby Robins Research Centre, The Institute of Cancer Research, London, UK. [2]Comprehensive Cancer Centre, School of Cancer & Pharmaceutical Sciences, King's College London, London, UK. [3]Breast Cancer Now Research Unit, Guy's Hospital, King's College London, London, UK. [4]The Royal Marsden NHS Foundation Trust, London, UK. [5]Oncology and Haematology Directorate, Guy's and St Thomas' NHS Foundation Trust, London, UK. ✉e-mail: clare.isacke@icr.ac.uk

however, to date no studies have interrogated BCLM tumour evolution to determine the timing of BCLM metastatic seeding. Exploring the molecular landscape and evolution of leptomeningeal malignancy is key to understanding what drives this disease and identifying targetable vulnerabilities.

Previous studies have demonstrated that CSF is an abundant source of tumour-derived cell-free DNA (cfDNA) in the setting of CNS-predominant malignancies, while plasma cfDNA is predominantly derived from extracranial sites[21,22]. Therefore we have utilised CSF cfDNA to interrogate the genomics and evolution of BCLM, and generated in vitro and in vivo models of BCLM by expansion of the scarce CSF disseminated tumour cells (DTCs). We demonstrate that BCLM shows early divergent evolution from the primary tumour and a distinct genomic profile compared to the primary tumour and other metastatic sites. Particularly noteworthy is the enrichment in BCLM of adherens junction and cytoskeletal genomic alterations, predicted to result in the acquisition of features associated with lobular breast cancers.

## Results

### Whole exome sequencing reveals unique genomic events in BCLM

Previously we described the prospective collection of CSF and plasma cfDNA from a cohort of BCLM patients, and demonstrated the utility of using ultra-low pass whole genome sequencing (ulpWGS) to measure the tumour-derived (ctDNA) fraction for BCLM diagnosis and therapy response monitoring[21]. Here we have investigated the genomic landscape of BCLM by whole exome sequencing (WES) in 21 patients for whom both matched CSF and plasma cfDNA were available along with archival samples of 18 matched primary tumour samples and 8 available metastatic site samples (Fig. 1a, b; Table 1; Supplementary Table 1). A threshold of ≥10% ctDNA fraction (assessed by ulpWGS[21]) was used to select cfDNA samples for whole exome sequencing (WES). As previously reported[21,22], when metastatic disease is restricted to the CNS and/or extracranial metastases are controlled, ctDNA content in plasma is low. Consequently although all 21 CSF cfDNA samples had ≥10% ctDNA fraction, only 11 of the plasma cfDNA samples met this criteria (Supplementary Fig. 1a). Plasma-derived cfDNA undergoing WES was sequenced to a higher coverage than CSF cfDNA and tumour tissues to mitigate for the lower ctDNA content (Supplementary Fig. 1b; Supplementary Table 2).

To assess the inter-tumoral heterogeneity between BCLM (CSF cfDNA), the primary tumour and extracranial metastases (plasma cfDNA or metastatic tissue), a multi-caller variant identification pipeline and post-hoc refinements approach was implemented to WES outputs to prioritise high-confidence mutations (see Methods; filtered non-silent somatic variants across all samples shown in Supplementary Data 1). Overall, tumour mutational burden (TMB) was as expected for the breast cancer genome[23], median CSF cfDNA 2.0, plasma cfDNA 2.0, primary tumour 1.3, metastasis 1.2 mutations/Mb with no significant difference between sample type (Supplementary Fig. 1c, d). Samples exhibiting high TMB are discussed later.

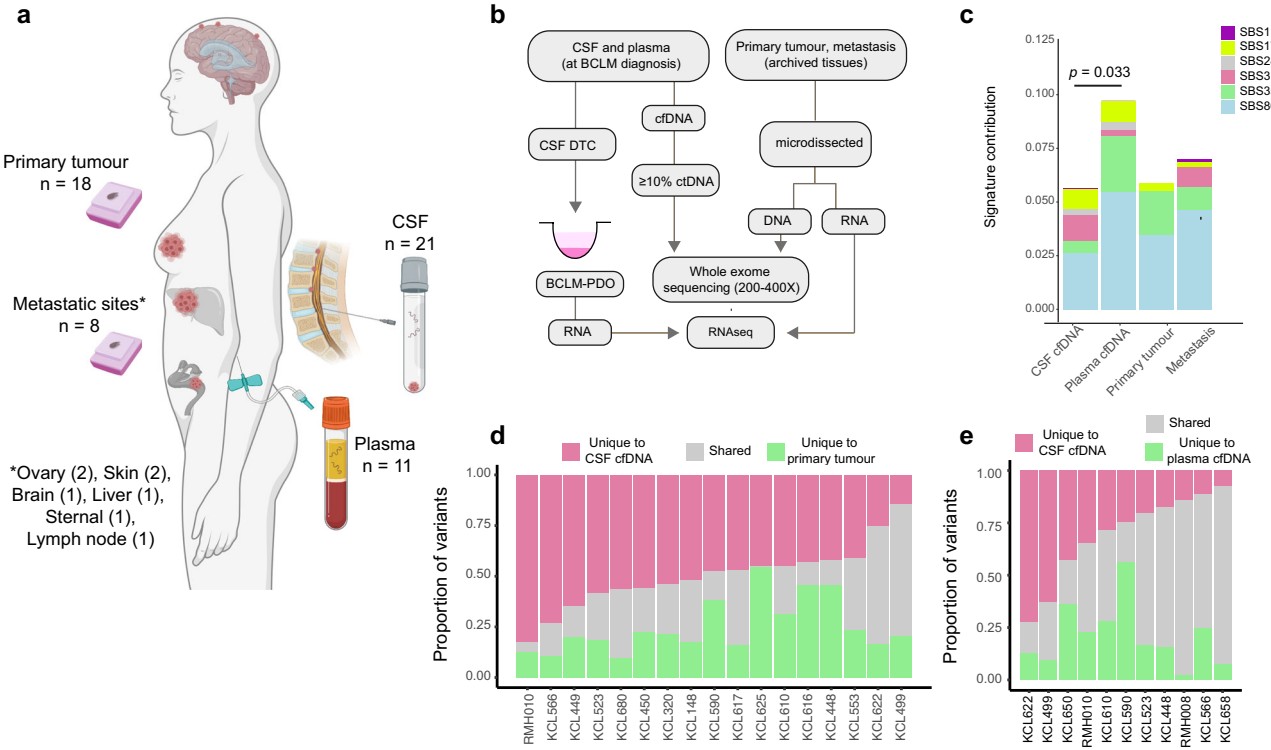

**Fig. 1 | Molecular profiling of BCLM by whole exome sequencing. a** Schematic showing the sites of material collected, created with BioRender.com. **b** Flowchart of sample processing pipeline. cfDNA, cell-free DNA; ctDNA, circulating tumour DNA; DTC, disseminated tumour cell; PDO, patient-derived organoid. **c**−**e** Samples underwent WES to identify mutation (exonic variants excluding synonymous SNVs, plus splice variants) and copy number aberration (CNA) events. **c** Chemotherapy-related COSMIC Single Base Substitution (SBS) mutational signatures compared between sites. Signature contributions of the six chemotherapy-related SBS signatures (SBS11, SBS17b, SBS28, SBS31, SBS35, SBS86) were compared between CSF cfDNA, plasma cfDNA, primary and metastasis tumour samples. Stacked bar plots display the mean signature contribution for each SBS signature across all samples in each category. Significantly different total chemotherapy-related SBS signature contribution between CSF and plasma cfDNA is shown (Mann-Whitney two-tailed test). All other comparisons were ns. **d** WES of paired CSF cfDNA samples collected at BCLM diagnosis and primary tumour samples (*n* = 17 pairs). Bars show proportion of total variants unique to CSF (median 47.8%; IQR 43.0−56.4%), unique to primary tumour (median 20.4%; IQR 16.6−31.5%) and shared (median 23.0%; IQR 13.8−33.7%). **e** WES of paired CSF cfDNA and plasma cfDNA samples collected at BCLM diagnosis (*n* = 11 pairs). Bars show proportion of total variants unique to CSF (median 24.6%; IQR 16.0−38.6%), unique to plasma (median 16.8%; IQR 11.3−26.6%) and shared (median 43.4%; IQR 24.2−65.1%).

**Table 1 | Clinicopathological characteristics of BCLM cohort**

| Study ID | Histological subtype | Time from mBC (years) | Parenchymal brain metastasis | Extracranial metastases | Extracranial disease status | BCLM survival (m) |
|---|---|---|---|---|---|---|
| KCL148 | Lobular | 4.7 | None | Liver | Stable | 5.71 |
| KCL320 | Lobular | 0.0 | None | None | Absent | 5.50 |
| KCL448 | IC-NST | 1.0 | Synchronous | Peritoneum, pericardium, distant nodes, soft tissue, bone | Progressive | 3.18 |
| KCL449 | Lobular | 0.6 | None | Bone | Stable | 27.43 |
| KCL450 | Lobular | 1.0 | None | Chest wall, peritoneal, pleura, bone | Progressive | 1.39 |
| KCL499 | IC-NST | 0.0 | Synchronous | None | Absent | 3.39 |
| KCL523 | Mixed | 1.2 | None | Peritoneum, bone, orbit | Stable | 6.04 |
| KCL553 | Lobular | 1.7 | None | Peritoneum, pleura, ovaries, bone, orbit | Stable | 18.39 |
| KCL566 | IC-NST | 1.1 | Metachronous (stable) | Distant nodes, peritoneum, subcutaneous, bone | Progressive | 0.86 |
| KCL590 | Micropapillary | 2.6 | None | Liver, spleen, bone | Progressive | 1.29 |
| KCL610 | Lobular | 6.5 | None | Bone | Unknown | 1.32 |
| KCL616 | Lobular | 2.9 | None | Bone, distant nodes | Unknown | 0.14 |
| KCL617 | Apocrine | 0.0 | None | Bone | Stable | 19.25 |
| KCL622 | Lobular | 1.3 | None | Bone, adrenal | Stable | 3.64 |
| KCL625 | IC-NST | 0.0 | None | None | Absent | 9.50 |
| KCL650 | IC-NST | 0.1 | None | Liver, bone | Progressive | 2.54 |
| KCL658 | IC-NST | 0.3 | Metachronous (stable) | Skin | Progressive | 9.75 |
| KCL680 | Lobular | 0.7 | None | Bone, peritoneum, ovaries | Stable | 25.61 |
| RMH008 | Lobular | 2.0 | Metachronous (stable) | Ovaries, cervix, vagina | Stable | 0.93 |
| RMH010 | Lobular | 4.0 | Metachronous (stable) | Peritoneum, liver, bone, orbit | Progressive | 3.39 |
| RMH011 | IC-NST | 5.6 | Metachronous (stable) | Lung, pleura, bone | Stable | 18.64 |

*BCLM* Breast cancer leptomeningeal metastasis, *cfDNA* cell-free DNA, *CSF* Cerebrospinal fluid, *IC-NST* Invasive carcinoma of no special type, *mBC* Metastatic breast cancer, *m* months, *NST* No special type, *PDO* Patient-derived organoids, *pBC* Primary breast cancer. Metachronous, prior to BCLM diagnosis; Synchronous, diagnosed concurrently with BCLM.

Mutational signature analysis was performed using the WES-identified single nucleotide variants (SNVs) (median of 96 filtered SNV per sample) (Supplementary Figs. 2–5; Supplementary Data 2 and 3). This displayed a higher contribution from chemotherapy-related single base substitution (SBS) signatures in plasma cfDNA compared to CSF cfDNA (Fig. 1c), which could reflect the 'sanctuary site' phenomenon whereby the CNS has a lower exposure to chemotherapy due to impedance by the blood-brain-barrier[24]. In keeping with prior findings in metastatic *vs.* primary breast cancer[20], CSF cfDNA samples had higher SBS13 (APOBEC) signature contribution than primary tumours, and lower SBS1 (clock-like/FFPE artefact) and SBS23 (unknown aetiology) signatures (Supplementary Data 3).

Indicative of a distinct genomic landscape in BCLM, 47.8% of mutations across CSF cfDNA and primary tumour were detected only in CSF cfDNA (Fig. 1d), with 20.4% detected only in primary tumour. Likewise, indicating a distinct mutational repertoire between BCLM and extracranial metastases, 24.6% of cfDNA mutations across the 11 paired CSF and plasma samples were detected only CSF cfDNA, with 16.8% detected only in plasma cfDNA (Fig. 1e). There was no significant correlation between the proportion of unique variants in CSF cfDNA *vs.* primary tumour or between CSF cfDNA *vs.* plasma cfDNA with time to development of BCLM nor to the number of previous treatment lines (Supplementary Fig. 6).

**BCLM subclones seed early from the primary tumour**

To investigate metastatic evolution of BCLM, variant allele frequencies and allele-specific copy number alterations (CNA; see below for CNA analysis) were used to infer mutational subclones in CSF and plasma cfDNA, primary tumour and metastatic tissues where sequenced. Clonal ordering and estimation of cancer cell fractions using ClonEvol allowed visualisation of BCLM metastatic evolution trees (Fig. 2;

Supplementary Figs. 7–13). Clonal modelling failed in one case (KCL625) due to the primary tumour and CSF cfDNA not sharing a common founder clone.

Within the limitation of WES, which may underestimate subclonal composition compared to whole genome sequencing, BCLM evolution analyses revealed: (a) BCLM metastatic seeding occurs early in primary tumour subclone evolution, predominantly at the stage of the founder clone (clone 1). In all 16 cases with primary tumour evaluated by clonal modelling, additional primary tumour subclone evolution occurred following BCLM metastatic divergence, and primary tumours were composed of 2 - 4 subclones. (Fig. 2a–c; Supplementary Figs. 7–10, 11c, 12a–c, 13c; Supplementary Table 3). (b) For the 11 cases with matched CSF and plasma cfDNA, divergent evolution between these occurred in 9 cases (Fig. 2a, b; Supplementary Figs. 7–8, 9a, 11b, 13b; Supplementary Table 3). Shared metastasis-specific branches were often seen, but CSF cfDNA, plasma cfDNA and other metastatic sites possessed distinct mutational subclones implying ongoing evolution at the leptomeningeal and extracranial sites. 2 cases showed linear co-evolution between CSF and plasma cfDNA (Supplementary Figs. 10a, 13a; Supplementary Table 3).

Phylogenetic tree construction allowed depiction of the metastatic seeding patterns in BCLM, with the limitation that not all metastatic sites were sampled. In the assessable cases, metastatic seeding was mostly monophyletic (Fig. 2a, b; Supplementary Figs. 7–8, 9a, 10a; Supplementary Table 3) where BCLM and other metastatic sites were derived from the same subclone within the primary tumour. However, one case showed polyphyletic seeding with metastatic sites originating from primary tumour at different timepoints in its evolution (Fig. 2c; Supplementary Table 3). As illustrated, KCL566 (Fig. 2a) displayed a monophyletic seeding pattern where metastases arose from the founding clone of the primary tumour (clone 1) and shared a common

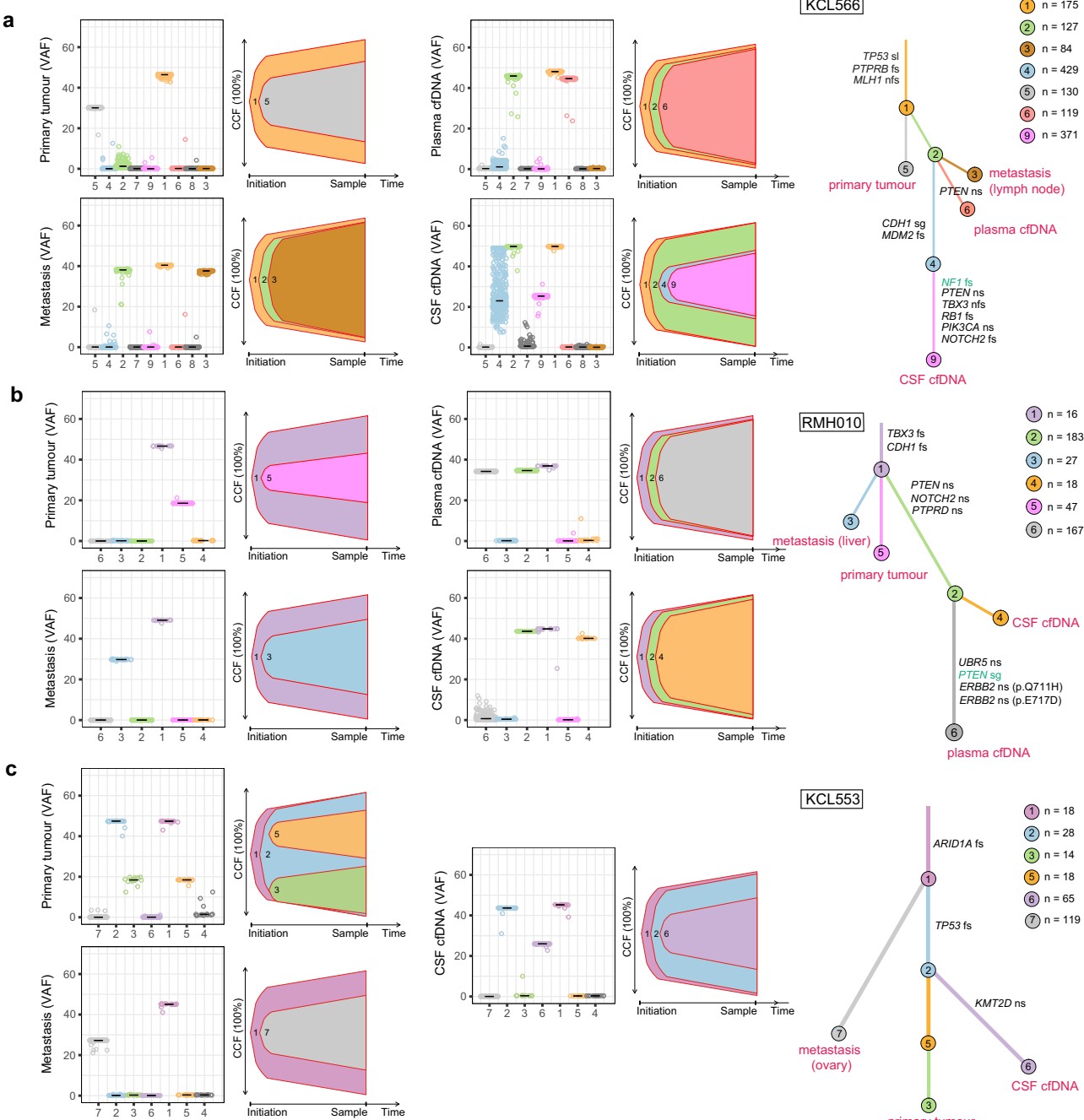

**Fig. 2 | BCLM clonal evolution modelling.** ClonEvol was used to infer consensus clonal evolution models, using the variant clusters generated by PyClone. (**a**) KCL566, (**b**) RMH010, (**c**) KCL553. Remaining models are presented in Supplementary Figs. 7–13. Box plots show the variant clusters identified in each sample, and displays the variant allele frequency (VAF) of each mutation, with the median VAF depicted by black lines. Bell plots show the ClonEvol estimated cancer cell fraction (CCF) and evolution order of clones within each sample. Phylogenetic trees for each case depicts the evolution pattern. Each branch represents the evolution path of a clone from its parental clone, and each node marks the clone number of the preceding branch. Clone colour and number match those in box and bell plots.

Samples are annotated at clone nodes when no further subclone evolution occurs. Branch lengths are scaled by the square root of number of mutations per clone, and branch width is inversely proportional to branch length. Mutations per clone are shown to top left of the trees. Gene names and variant type are annotated on branches as follows; black font if appear in the list of BCLM frequently altered cancer driver genes (shown in Fig. 3); green font if variant is OncoKB annotated as potentially therapeutically actionable (Supplementary Table 5). Variant type abbreviations; fs frameshift, nfs non-frameshift, ns nonsynonymous, sg stop-gain, spl splice-site variant.

evolution branch (clone 2) before displaying divergent evolution between lymph node metastasis, plasma and CSF cfDNA. At the time of CSF and plasma sampling, progressive disease was present at extracranial sites (peritoneum, subcutaneous tissues and bone), potentially driven by *PTEN* p.L23S mutation, while CSF cfDNA depicted significant further parallel evolution, with two further subclones of high mutation

count (clones 4 and 9; 429 and 371 mutations, respectively), displaying a *CDH1* truncating mutation p.L139X and *NF1* frameshift mutation. RMH010 (Fig. 2b) shows early monophyletic seeding of all metastases from the primary tumour (clone 1), with divergent evolution between liver metastases (6 years post primary diagnosis) and CSF and plasma cfDNA (10 years post primary diagnosis, progressive disease in

leptomeninges, peritoneum, liver, bone and orbital soft tissues) implying ongoing tumour evolution at metastatic sites. KCL553 (Fig. 2c) showed a polyphyletic pattern of metastatic seeding with ovarian metastasis seeding from the founding clone of the primary tumour containing *ARID1A* frameshift mutation (clone 1) and BCLM (CSF cfDNA) seeding after a clonal sweep in the primary tumour comprising *TP53* frameshift mutation (clone 2). Further subclone evolution in the CSF cfDNA was apparent with development of a *KMT2D* missense variant.

7/21 BCLM cases had co-existing brain parenchymal metastasis (Table 1), the majority developing metachronous to BCLM. Archived brain metastasis tissue, resected 3 years prior to BCLM development, was available for RMH011 (Supplementary Fig. 11a). Clonal modelling revealed that BCLM and brain metastasis shared two common ancestral clones, however displayed divergent evolution with unique subclones at each site. No primary tumour was available for sequencing to determine the metastatic seeding pattern of this case.

### Cancer driver gene aberrations enriched in BCLM

We identified 35 cancer driver genes (defined in Methods) which were frequently aberrated in CSF cfDNA by non-silent somatic mutation or high-level copy number change (deep deletion or amplification) (Fig. 3; Supplementary Fig. 14). Driver gene aberrations occurring exclusively in CSF cfDNA and not shared with primary/metastatic tumour or plasma cfDNA (where sequenced) were found in genes involved in histone modification (*KAT6B, KMT2D, NUTM2A*) and microtubule formation (*PDE4DIP*). Frequently aberrated cancer driver genes in plasma cfDNA are shown in Supplementary Table 4, however, given the small number of plasma cfDNA samples sequenced, the identification of extracranial metastasis driver genes is better represented by larger published cohorts[16,18,25].

Next, we examined the alteration rate of the BCLM altered genes to a publicly available sequencing dataset of 216 non-BCLM metastatic breast cancer samples[26], herein known as MBC cohort (Fig. 3; Supplementary Fig. 14). This comparison revealed 10 genes, *CDH1, MUC16, TBX3, ARHGEF10, MDM2, WRN, CTNNA1, NUTM2A, PSIP1* and *MLH1*, with a significantly different alteration rate in BCLM compared to the MBC cohort ($p \leq 0.05$, Chi-square test) (Supplementary Data 4).

### Copy number differences between BCLM and matched samples

Copy number alteration (CNA) across the genome was determined using WES data, revealing common breast cancer events across samples such as 1q and 8q gains, and 8p and 16q losses[27] (Fig. 4a; Supplementary Data 5). Based on the proportion of CNA events across the exome in each sample type, we identified genomic loci that differ in CNA frequency between CSF cfDNA and matched primary tumours and between CSF and matched plasma cfDNA samples (Fig. 4b; Supplementary Fig. 15). Compared to primary tumours, CSF cfDNA more frequently lost 13q regions containing *PCDH9* and *PCDH17* (members of the cadherin superfamily) and *KLHL1*, an actin binding protein with a role in cytoskeleton reorganisation. Broad 16q regions containing the classical cadherin *CDH1* and other cadherins showed more frequent loss in CSF than plasma cfDNA samples. Gained regions in CSF cfDNA compared to plasma included 1q21.3 containing the intermediate filament-associated proteins *TCHH* and *FLG2*, and S100 family genes which have been implicated in poor prognosis breast cancer relapse[28].

### Gene set enrichment of recurrently altered genes in BCLM

The ability of tumour cells to gain access to the leptomeningeal space via the blood-brain/blood-CSFbarrier, and successfully interact with the unique microenvironment in this space, may require genomic alterations not currently characterised as cancer driver genes. Consequently, we took an unbiased approach of performing gene set enrichment analysis of all frequently altered (by non-silent mutation, amplification or deep deletion) genes in CSF or plasma cfDNA

compared to primary tumours (Fig. 4c, d; Supplementary Fig. 16). CSF and plasma cfDNA samples shared alterations in the gene ontology (GO) terms 'endomembrane system organisation', and 'fibroblast growth factor response'. By contrast, GO terms showing enrichment in CSF cfDNA compared to primary tumours but not in plasma cfDNA were; 'positive chemotaxis', 'myelination' and 'supramolecular fiber organization', the latter encompassing key cytoskeletal scaffold components such as actin filaments, intermediate filaments and microtubules, and their regulators. Within the 'supramolecular fiber organization' term, *CTNNA1, MYO15A, DMTN* and *SPTA1* are of interest due their role in regulating adherens junctions and the actin cytoskeleton. Within 'myelination', CSF cfDNA samples were enriched for genomic alterations in *ARGHEF10* (discussed above), and genes with roles in cell-cell adhesion including neurofascin (*NFASC*) and teneurin transmembrane protein 4 (*TENM4*). Within 'positive chemotaxis', amplifications were found in scribble planar cell polarity protein (*SCRIB*), previously demonstrated to promote cell migration and metastasis[29–31]. Genomic alteration in beta-defensin genes (*DEFB4A, DEFB103B, DEFB104B*), antimicrobial peptides with multiple proposed functions including regulation of cancer cell migration[32], were also enriched in BCLM samples.

### Enrichment of adherens junction components and cytoskeletal aberrations in BCLM

The high rate of *CDH1* (E-cadherin) mutations (52%) in BCLM CSF is consistent with the enrichment of ILCs in this cohort since deleterious *CDH1* mutations are an early driver event in the majority (63%) of ILCs, leading to defective adherens junctions[12,13]. *CDH1* aberrations are rare in non-lobular breast cancers (2.3% TCGA IC-NST cases). Mutations and copy number aberrations in *CTNNA1* (α-catenin), an indispensable adherens junction component which links the cadherin/β-catenin complex to the underlying actin cytoskeleton, has been suggested as an alternative mechanism to becoming 'lobular'[33], however *CTNNA1* genomic aberrations are rare in both ILC and non-lobular breast cancers (0.5% and 1.5% respectively, TCGA). In our BCLM cohort, we identified deleterious alterations of *CDH1* or *CTNNA1* in 55% of non-lobular BCLM cases (Fig. 3; Supplementary Fig. 14). For example, KCL566 (IC-NST) harboured a stop-gain *CDH1* mutation plus *CDH1* LOH in CSF cfDNA, with neither aberration being detected in the matched plasma, primary tumour or metastatic lymph node samples. KCL617, arising from a breast cancer of apocrine morphology, displayed a frameshift *CDH1* mutation in both primary tumour and CSF cfDNA, with additional *CDH1* loss of heterozygosity (LOH) in the CSF cfDNA. KCL622 and KCL625 (both IC-NST) displayed *CTNNA1* focal deep deletions in CSF cfDNA, while *CTNNA1* had neutral copy number status in matched primary tumours. KCL658 (IC-NST) displayed a *CTNNA1* stop-gain mutation in CSF cfDNA, shared with the matched plasma cfDNA and KCL650 (IC-NST) displayed a *CTNNA1* focal deep deletion in CSF cfDNA, while the matched plasma cfDNA showed shallow copy number loss only. For KCL658 and KCL650, however, no primary tumour was available for sequencing to determine the timing of *CTNNA1* aberration. *CDH1* and *CTNNA1* aberrations were often BCLM unique events and mutually exclusive ($p = 0.0352$; two-sided Fisher's exact test) implying that adherens junction defects occur during leptomeningeal metastasis evolution.

In addition to the adherens junction components *CDH1* and *CTNNA1*, sequencing analysis also revealed aberrations in cytoskeletal components and their regulators (Fig. 4d; Supplementary Fig. 16). For example, a member of the RhoGEF family, *ARHGEF10* was frequently aberrated in BCLM CSF cfDNA (23% of cases), comprising four deep deletions (one focal event ~1.9 Mb and 3 broader, but not arm-level, events of between 6.7 and 6.9 Mb) (Supplementary Data 5) and a BCLM unique missense mutation (p.A1100P) with predicted pathogenicity by SIFT, Polyphen2 and CADD prediction scores (Supplementary Data 1). A further, nine CSF cfDNA samples demonstrated *ARHGEF10*

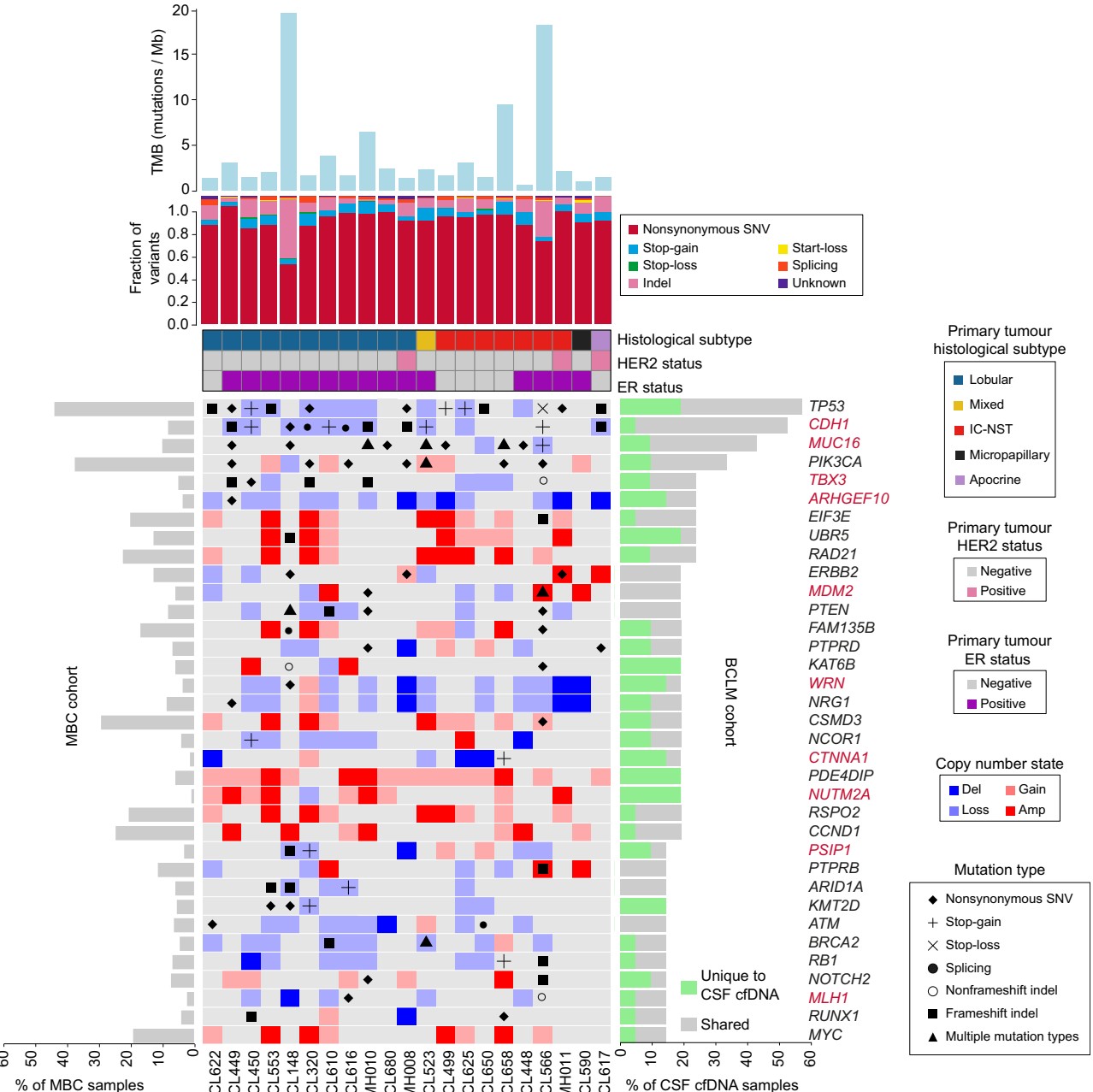

**Fig. 3 | Cancer driver gene aberration in BCLM CSF cfDNA samples.** Uppermost bar plot show tumour mutational burden (TMB) by sample. Stacked bar plot below shows mutational subtype for each sample as fraction of all variants (excluding synonymous). Banners indicate primary tumour histological subtype and ER and HER2 status. Main plot shows the 35 most frequently altered cancer driver genes (alterations include non-silent somatic mutation or high-level copy number change) in CSF cfDNA as follows; Cancer Gene Census (CGC) gene altered in at least 4 cases; breast cancer driver gene (defined in Methods) altered in at least 3 cases. Right side bar plot shows percentage of CSF cfDNA samples ($n = 21$) with gene

alterations that are unique to the CSF sample (green) or shared with any other site per individual (grey). Left side bar plot shows genomic alteration rate in the MBC cohort. See Supplementary Fig. 14 for extended data on matched plasma cfDNA, primary tumour and metastasis samples, and Supplementary Table 4 for frequently altered cancer driver genes in plasma cfDNA. Gene names are highlighted in red if genomic alteration rate was significantly different ($p \le 0.05$ by Chi-square test two-sided) in BCLM cohort compared to a publicly available dataset of 216 metastatic breast cancer (MBC cohort) samples (see Methods). Extended data in Supplementary Data 4 displays the significance values for this comparison.

heterozygous losses. *ARHGEF10* facilitates activation of Rho and downstream Rho kinase (ROCK) to promote actomyosin contractility[34]. Excessive actomyosin contractility has been reported to impair adhesion to, and migration on, collagen I and is deleterious to ILC development[35], consequently it is tempting to speculate that loss of *ARGHEF10* provides a pro-survival mechanism for BCLM. Myosin 15 A (*MYO15A*), encoding an actin-based motor molecule, was aberrated in 6/21 CSF cfDNA samples (one truncating mutation and four missense mutations with predicted pathogenicity), often

accompanied by loss-of-heterozygosity (LOH). Dematin actin binding protein (*DMTN*), a regulator of cytoskeleton remodelling[36,37], was frequently aberrated in CSF cfDNA. *DMTN* downregulation has been reported to promote colorectal cancer metastasis through activation of Rac1, a key cytoskeletal regulator[38]. Although the *DMTN* copy number events were part of broader 8p deletion events (18–29 Mb), frameshift and missense mutations were found in 3 CSF cfDNA samples and were predominantly BCLM unique mutations. Finally, spectrin-alpha 1 (*SPTA1*), a scaffold protein linking the plasma membrane to the

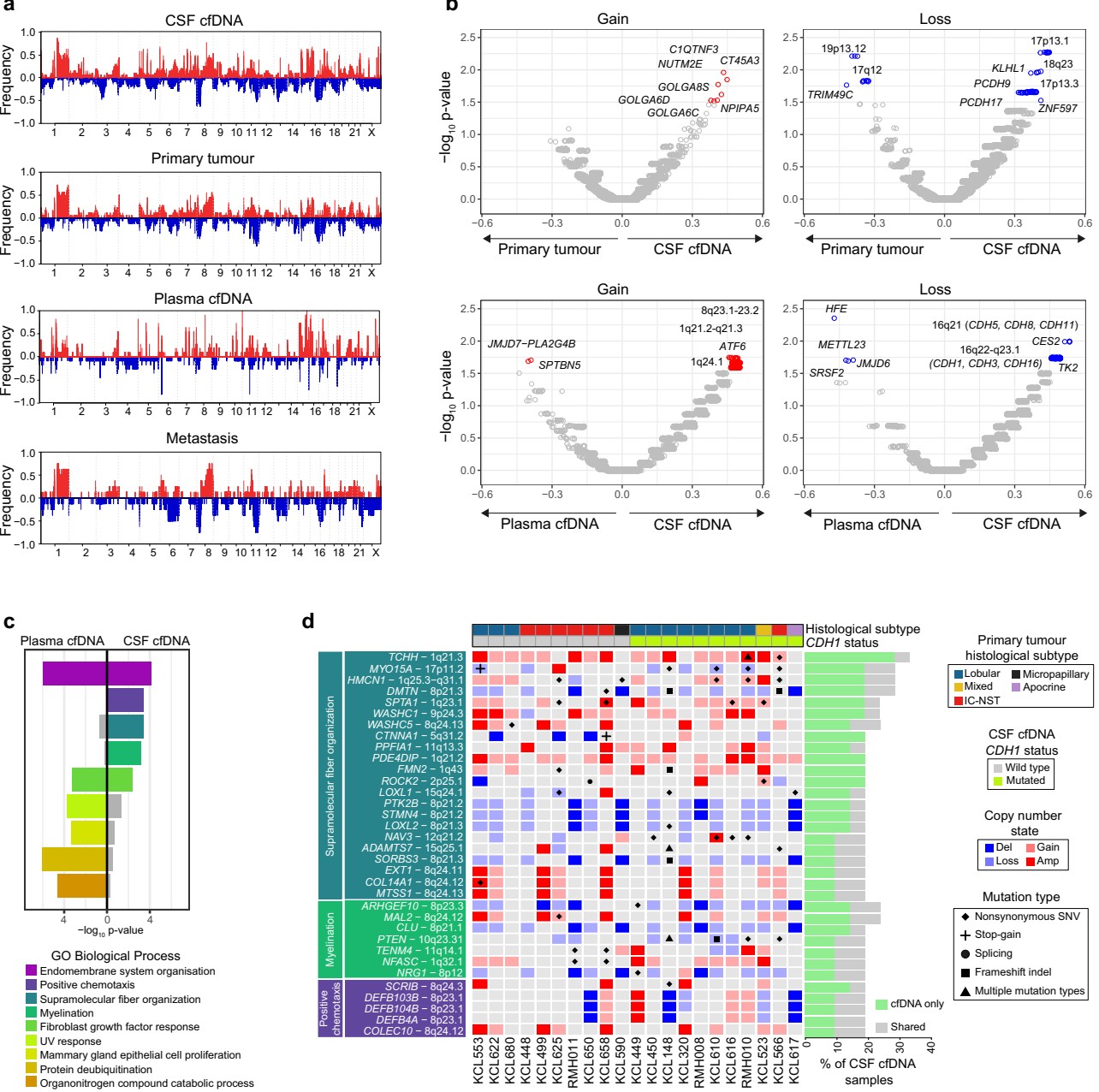

**Fig. 4 | BCLM CNA landscape and gene set enrichment. a** Genome wide purity-adjusted CNA plots of all samples (CSF cfDNA (*n* = 21), plasma cfDNA (*n* = 11), primary tumour (*n* = 18) and metastasis (*n* = 8)) showing frequency of copy number gain (red bars, positive y-axis) and copy number loss (blue bars, negative y-axis) at a gene level. **b** Volcano plots displaying comparison of CNA frequency per gene, between CSF cfDNA and primary tumours (upper panels) and between CSF cfDNA and plasma cfDNA (lower panels). Delta proportion (y-axis) is the proportional difference in CNA frequency between sample types. Significance (x-axis, -log$_{10}$ *p*-value) obtained by Chi-squared proportion test (two-sided) compares CNA event proportion per gene between sample types (source data are provided as a Source Data file). Gene names are annotated if delta proportion is ≥ ± 0.20 and -log$_{10}$ *p*-value ≥ 1.5 (as shown in Supplementary Fig. 15). **c** Gene set enrichment analysis. Genes enriched for alterations (including non-silent somatic mutation or high-level copy number change) in CSF cfDNA (*n* = 257) and/or plasma cfDNA (*n* = 160) *vs.*

matched primary tumour, underwent statistical overrepresentation test using GO biological process (PANTHER). Displayed terms were retained if -log$_{10}$ *p*-value was ≥ 3.0 in either sample type (Fisher's exact test, adjusted for multiple testing using the Benjamini-Hochberg procedure with an overall false discovery rate of < 0.05). **d** Focused alteration plot of genes from the indicated GO biological processes overrepresented in CSF cfDNA *vs.* matched primary tumour, but not in plasma cfDNA *vs.* primary tumour. Banners above indicate; histological subtype (primary tumour) and *CDH1* mutation status of CSF cfDNA. Left panel shows gene names and chromosomal location, grouped by GO biological processes. Bars to right show the frequency of gene alteration across CSF cfDNA samples (*n* = 21) with 'cfDNA only' indicating alterations unique to CSF cfDNA or CSF and plasma cfDNA at the time of BCLM diagnosis, and 'shared' occurring both in CSF cfDNA and primary tumour or metastasis sample. See Supplementary Fig. 16 for extended data on matched plasma cfDNA, primary tumour and metastasis samples.

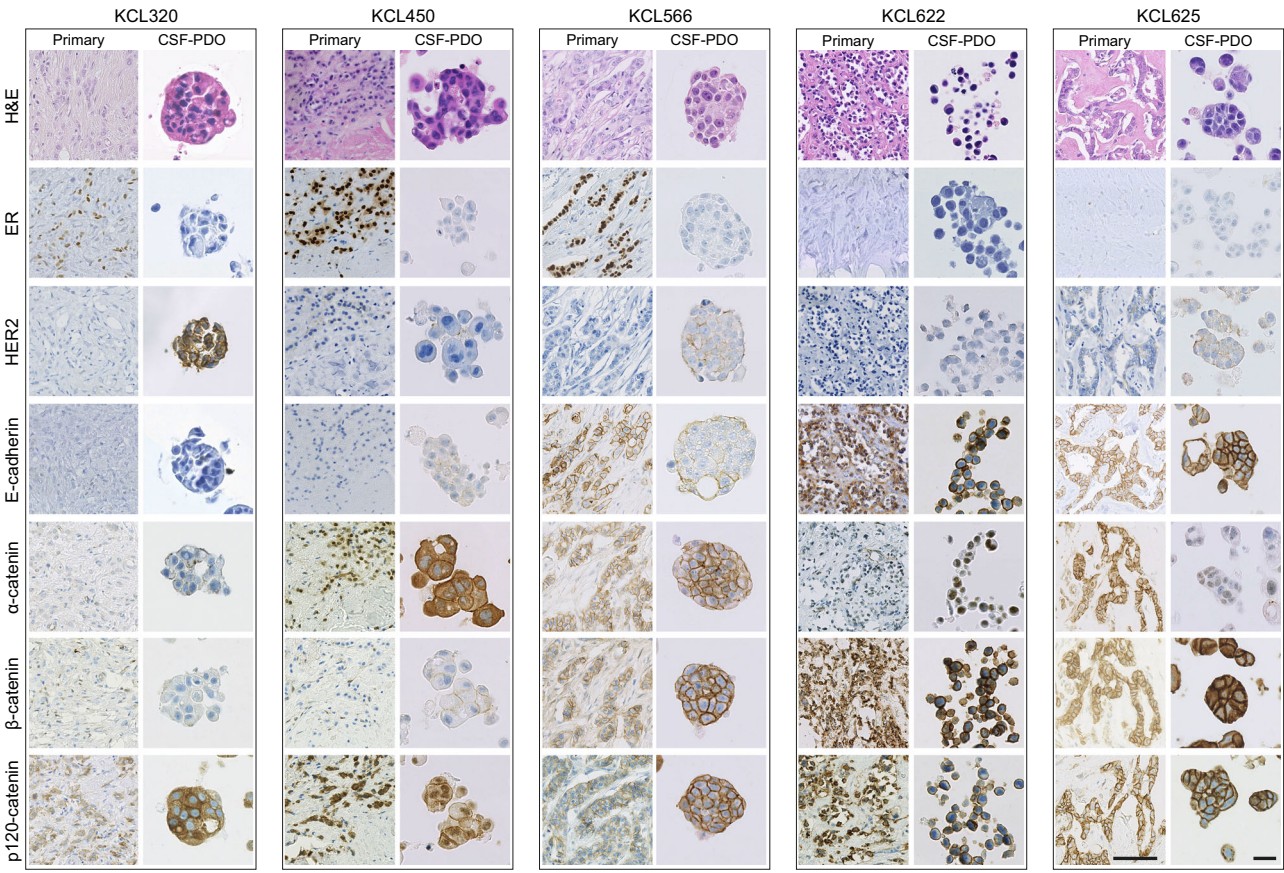

**Fig. 5 | Generation of BCLM patient-derived organoids (PDOs) established from CSF DTCs.** Histology and immunohistochemistry of BCLM PDOs and matched primary tumours. A minimum of 2 biological replicates were performed for each PDO immunohistochemical staining (range 2–4 replicates) with representative images shown. Scale bars, 50 μm.

actin cytoskeleton, was aberrated in 4/21 BCLM CSF cfDNA samples by somatic missense mutations, of which three had high pathogenicity predictions.

## Actionable mutations discovered by CSF cfDNA sequencing

Actionable variants (OncoKB therapeutic alterations level 1–4) were identified in 17/21 (81%) CSF cfDNA samples, and were private to CSF in 4/21 (19%) of cases (Supplementary Table 5, with potentially actionable variants highlighted in green in Fig. 2 and Supplementary Figs. 7–13). Recurrent actionable variants across the cohort were *PIK3CA* activating mutations ($n = 5$), *ARID1A* deleterious mutations ($n = 3$), *MDM2* amplifications ($n = 3$) and *PTEN* deleterious mutations ($n = 2$). Variants emergent in CSF which were not found in primary tumours were: a *BRCA2* frameshift mutation p.N1599Ifs*18 accompanied by LOH in KCL610 predictive of response to PARP inhibitors; a mono-allelic *BRCA1* frameshift mutation p.Y655Vfs*18 in KCL148 with more speculative prediction of response to PARP inhibition; an *NF1* frameshift mutation p.Y1586Ifs*17 in a late BCLM-subclone of KCL566 potentially targetable by MEK inhibition; and an *ESR1* p.Y537 mutation private to KCL680 BCLM subclones and predictive of a loss of response to aromatase inhibitor but retaining sensitivity to the oestrogen receptor (ER) degrader, fulvestrant. *ERBB2* amplification/gain in CSF cfDNA were found only in the 3 known HER2+ cases. In 2 of these *ERBB2* mutations were also found; p.L755S in RMH008 (actionable) and p.L800F in RMH011 (a kinase domain mutation of unknown actionability). A further *ERBB2* variant (p.V597M, unknown actionability) was detected in KCL148, without *ERBB2* amplification. All *ERBB2* mutations were early clonal events (Fig. 2, Supplementary Figs. 11a, b, 12a), as were all actionable *PIK3CA* mutations.

Of note, two CSF samples, KCL566 and KCL148, exhibited high TMB (>10 mutations/Mb) (Fig. 3; Supplementary Fig. 1d). Both these cases had truncal aberration in the mismatch repair gene *MLH1* and high primary tumour TMB. However, together with KCL658, both showed substantial further elevation of TMB in CSF cfDNA, indicating continuing accumulation of mutational events during evolution in the leptomeningeal space. High TMB presents an actionability target through prediction of response to immune checkpoint inhibition[39].

## BCLM patient-derived organoids (PDOs)

A major challenge in developing preclinical models of BCLM is the lack of access to clinical biopsies. To overcome this challenge, we optimised methodologies for in vitro expansion of the small number of viable DTCs present in CSF samples. This included miniaturising the initial culture conditions and inclusion of meningeal cell conditioned medium during the early PDO culture (see Methods for full details), resulting in the successful development of 5 BCLM PDOs grown and passaged as Matrigel-embedded cultures (Fig. 5). Of note, although it has been questioned whether DTCs within the CSF represent the entire leptomeningeal metastatic population, Remsik and colleagues[40] have recently reported a high degree of plasticity between 'floating' and adherent tumour cells in the CSF with the floating cells displaying a more aggressive phenotype. The BCLM PDOs were subjected to WES and a comparison of CNAs with their matched CSF cfDNA samples revealed a high level of concordance (Fig. 6a upper panels). Similarly, comparison of the mutational landscape revealed a median of 81.5% of CSF cfDNA detected variants were present in their matched BCLM PDOs (Fig. 6a lower panels). Comparison of BCLM PDOs and primary tumours revealed a lower concordance in both CNA and mutations

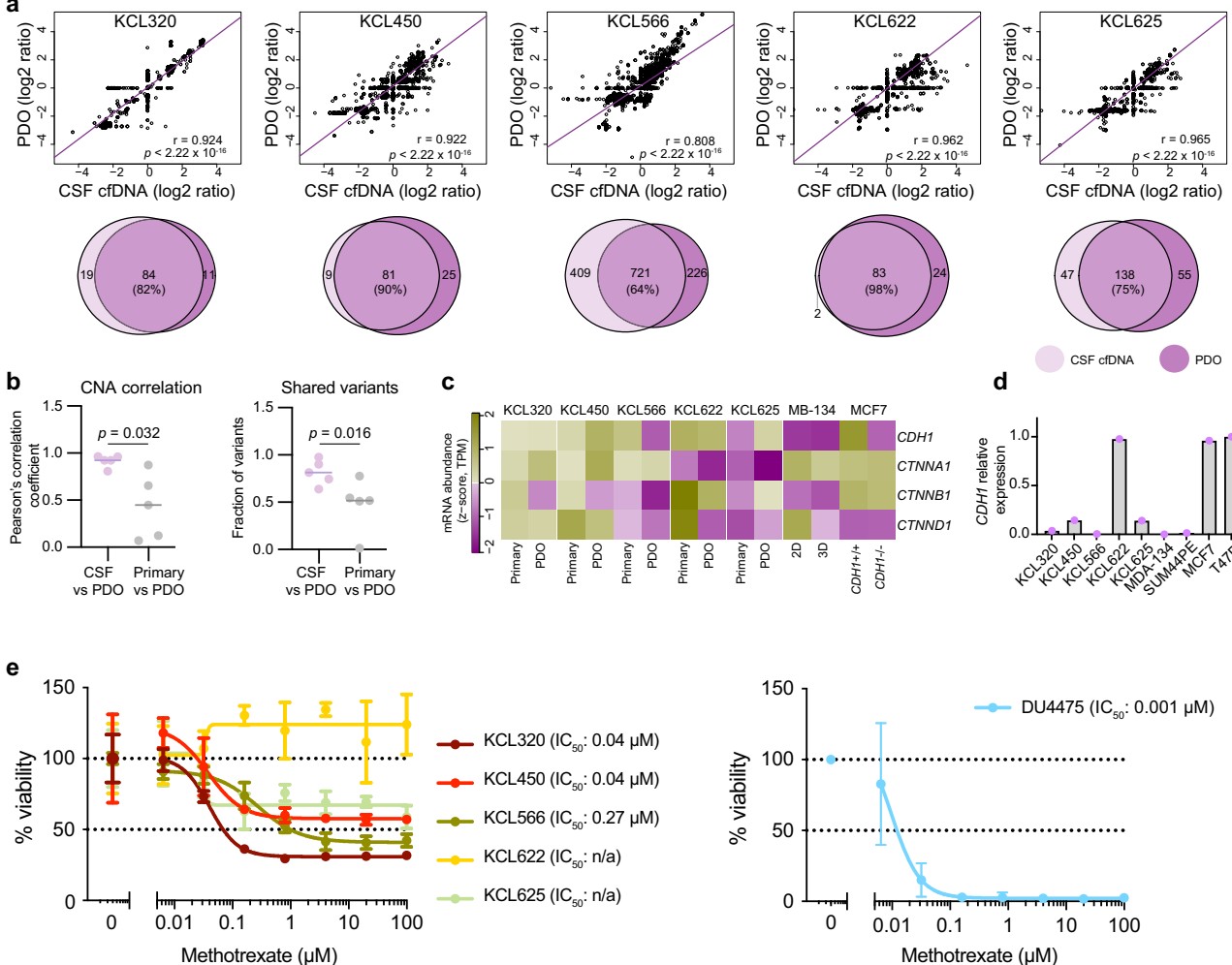

**Fig. 6 | Characterisation of BCLM PDOs. a** Genomic comparison of CSF cfDNA and matched BCLM PDO (early passage) by WES. Top panel, correlation plots of copy number Z-score (one data point per gene) with regression line plotted (purple) and Pearson's r-value plus two-sided significance values are shown. Lower panel, Venn diagrams of somatic mutations (excluding synonymous) present in CSF cfDNA and matched BCLM PDO, values shown are mutation count. **b** Comparison of Pearson r-values from Fig. 6a and Supplementary Fig. 17 by Mann-Whitney, two-tailed test showing a higher level of correlation in CNA status between PDO ($n = 5$) and CSF cfDNA ($n = 5$) than between PDO and primary tumour DNA ($n = 5$) (left panel) and higher fraction of shared variants between CSF cfDNA and PDO than primary

tumour and PDO (right panel), source data are provided as a Source Data file. **c** RNAseq of PDOs (fresh early passage) and matched primary tumours (FFPE). Expression levels of adherens junction components are shown as Z-scores of log2-TPM values (transcripts per million). **d** RTqPCR analysis of *CDH1* expression in PDOs as compared with cell line controls. $n = 1$ biological replicate, $n = 3$ technical replicates, source data are provided as a Source Data file. **e** BCLM PDOs and cell line control (DU4475) were cultured in the presence of methotrexate in 0.2% DMSO or 0.2% DMSO alone (vehicle alone). Cell viability was measured by CellTiter-Glo after 14 days. Data represents mean of $n = 4$ wells per datapoint ± SD, normalised to vehicle alone treatment, source data are provided as a Source Data file.

(Fig. 6b; Supplementary Fig. 17). Overall these data indicate that CSF cfDNA and DTCs represent a valid surrogate for the rarely available BCLM material.

Immunohistochemical staining of PDOs and matched primary tumours (Fig. 5) revealed a reduction of E-cadherin levels in PDO KCL566 (IC-NST with acquired *CDH1* truncating mutation in CSF cfDNA and BCLM PDO), commensurate with the reduced *CDH1* expression compared to its primary tumour (RNAseq; Fig. 6c), the lack of PDO *CDH1* expression detected by RTqPCR (Fig. 6d) and the lack of E-cadherin staining when grown as patient-derived xenografts (PDXs; see later). Similarly, there was a loss of membrane associated α-catenin staining and reduced *CTNNA1* expression in PDO KCL622 and PDO KCL625, both IC-NST showing acquired *CTNNA1* deep deletions in CSF cfDNA and BCLM PDO. p120-catenin in a cytoplasmic, as opposed to membranous, location is a hallmark of classical ILC[41] and was observed in the primary ILC tumours and PDOs KCL320 and KCL450 (Fig. 5). By contrast, PDO KCL566 showed mixed cytoplasmic and membranous p120-catenin and PDOs

KCL622 and KCL625 showed membranous p120-catenin. These findings agree with published reports, that *CTNNA1* loss in the presence of *CDH1* expression, does not lead to p120-catenin cytoplasmic localisation despite *CTNNA1* deficient cells acquiring 'lobular-like' features of a rounded cell morphology and anoikis resistance[42].

We noted that the 3 PDOs derived from patients with ER+ primary tumours (KCL450, KCL566, KCL320) displayed loss of ER expression (Fig. 5; Supplementary Table 6). Although loss of ER expression is found in ~25% of PDO derived from ER+ tumour samples, likely secondary to establishing PDO culture, reduced ER expression in metastases from ER+ primary breast cancers is also reported[43–45], and linked to more frequent *TP53* mutation in endocrine therapy-resistant metastatic diease[46]. Our BCLM samples showed higher rates of *TP53* mutation and copy number loss in CSF cfDNA (12/21) than the primary tumour (6/18) (Supplementary Fig. 14).

Finally, immunohistochemical staining revealed enhanced HER2/ERBB2 levels in 4/5 PDO (ranging from 1+ to 3+ on IHC scoring), where

in all cases the matched primary tumours were HER2 negative (0 on IHC scoring) (Fig. 5; Supplementary Table 6).

The standard-of-care treatment for patients with BCLM is delivery of methotrexate into CSF, intrathecally via repeated lumbar puncture or an intraventricular device such as an Ommaya reservoir, however randomised trial evidence for its effectiveness is lacking[47]. Compared to a breast cancer cell line (DU4475, grown as 3D spheroids in PDO culture conditions) which shows complete response to methotrexate, PDO KCL622 shows no response and the remaining 4 BCLM PDOs show only a partial response despite the high drug concentrations used (Fig. 6e).

**PDX models reveal distinct BCLM organotropism**

Although informative, to date mouse models of BCLM have been limited to selecting non-BCLM-derived breast cancer cell lines for their ability to grow in the leptomeninges[48]. To study the organotropism of CSF DTC-derived PDOs, PDOs were transduced to express mCherry and luciferase 2 (mChLuc2) (Supplementary Fig. 18) prior to inoculation into immunocompromised mice. Orthotopic inoculation into the mammary fat pad revealed a variable ability of BCLM PDOs to form primary breast tumour xenografts (Fig. 7a). For PDOs KCL320 and KCL450 only small tumour masses were detectable in a subset of inoculated animals, with no evidence of spontaneous metastasis to secondary sites. Of note, these PDOs were derived from the two patients with luminal, de novo ILCs (KCL320 and KCL450). By contrast, PDOs KCL566, KCL622 and KCL625 readily formed primary tumours with KCL566 and KCL622 showing widespread spontaneous metastasis including, in a subset of mice, to the leptomeninges. PDO KCL320 is of interest as it is derived from CSF DTCs of a patient with metastatic disease restricted to the leptomeninges, and in vivo showed preference for growth in the CNS environment. In addition to its failure to grow robustly after orthotopic inoculation (Fig. 7a), PDO KCL320 did not grow following intraductal inoculation into the mouse mammary gland (MIND model) (Fig. 7b) or by intraperitoneal inoculation (Fig. 7c). However, following intracardiac inoculation, 4 out of 5 BCLM PDOs, including PDO KCL320, resulted in colonisation of the leptomeninges and/or the brain parenchyma (Fig. 8a), and intracerebroventricular (ICV) inoculation of KCL320 PDO directly into the CSF resulted in two out of the four mice developing metastatic disease restricted to the leptomeninges (Fig. 8b). In addition to the leptomeninges, BCLM PDOs display a strong predilection for metastasising to endocrine organs such as the ovaries, adrenal glands and pituitary gland, a feature associated with the metastatic spread of ILC[49]. This pattern of metastatic spread in the mice was distinct from the metastatic involvement in the patients from whom the PDO cohort was derived, with only 1/5 patients (KCL625) recorded to have metastatic disease in endocrine tissues (Table 1). The in vivo endocrine tissue organotropism likely reflects the enrichment for alterations in E-cadherin, the associated junctional component α-catenin and/or in cytoskeletal regulators during BCLM evolution, indicating that colonisation of the leptomeninges strongly selects for lobular-like features.

## Discussion

Recent studies in metastatic breast and other solid tumours, have demonstrated specific genomic alterations in metastases beyond those observed in primary tumours[15–20]. However, to date, metastasis to the leptomeninges has been relatively unstudied due to difficulty accessing material from this rarely biopsied site. Here we present genomic profiling of a cohort of BCLM samples (both CSF cfDNA and CSFDTCs expanded in culture as PDOs) with concurrently sequenced plasma cfDNA and archival tumour tissues, allowing comparison of BCLM with the primary tumour and extracranial metastatic sites. Through phylogenetic modelling we determined that BCLM exhibits early clonal divergence from the primary tumour, and that metastases spatially separated by the blood-brain/blood-CSFbarrier bear distinct

genomic changes, indicating that BCLM evolve biological features favourable for tropism to and/or survival within the leptomeningeal niche. It was also revealed that leptomeningeal and extracranial metastases have different evolutionary pressures, with plasma cfDNA bearing increased marks from previous chemotherapy treatment compared to CSF cfDNA, implying that the blood-brain/blood-CSF-barrier may effectively create a sanctuary to CNS disease.

Sequencing CSF cfDNA uncovered additional genomic alterations with predicted or potential actionability over primary tumour sequencing alone in 19% of cases. This finding supports the wider prospective assessment of the clinical utility of CSF liquid biopsy for precision medicine approaches in BCLM, however important caveats to interpreting the potential actionability of these identified variants are (a) the uncertainty of drug penetration through the blood-brain and blood-CSF barriers to reach therapeutic concentrations within the CSF, and (b) limited licensed indications for agents that target the biology associated with these variants in a breast cancer setting. High TMB was discovered in 2 cases with substantial elevation of TMB in the CSF cfDNA setting in 3 cases; a finding that has therapeutic relevance given that high TMB is predictive of response to immune checkpoint inhibitors[39], and recent studies showing promising clinical efficacy of pembrolizumab[50] and that, despite being administered intravenously, pembrolizumab increases the number of cytotoxic T-cells and IFN-γ signalling in CSF[51]. CSF DTCs expanded into PDO culture allowed for immunohistochemical profiling and revealed changes in receptor levels, such as increased HER2 compared to the primary tumour. We did not identify copy number level changes in *ERBB2*, in agreement with published cohorts in breast cancer brain metastasis which show no amplification of *ERBB2* but increased *ERBB2* expression in brain metastases compared to their matched primary tumours[52,53]. The increase in HER2 levels in non-amplified (HER2-low) cancers has therapeutic implications for CNS disease given the efficacy of systemically-delivered HER2-ADCs in both HER2-amplified and HER2-low brain metastases, with trials ongoing in BCLM[54,55].

A striking finding relevant to BCLM biology was the enrichment for alteration of genes involved in cell-cell adhesion, cytoskeletal regulation and cell migration. It is well established that ILCs have a distinct biology, defined by a lack of cell-cell adhesion as a result of inactivation of *CDH1*, displaying a discohesive and migratory phenotype[56,57] and a metastatic site predilection to leptomeninges and other fluid-filled serosal-lined cavities including peritoneum, pleural and pericardial spaces[58]. We found that BCLM developing in non-ILC cases, was associated with ILC-like genomic alterations such as *CDH1*, *CTNNA1* and *ARHGEF10* loss-of-function alterations. *CTNNA1* is a key component of the E-cadherin adherens junction while *ARHGEF10* is a GTP exchange factor for RhoGTPases with in vitro studies demonstrating increased cell motility and invasiveness in *ARHGEF10* depleted pancreatic cancer cells[59]. We propose that *CTNNA1* and *ARHGEF10* defects may be an alternative mechanism to become 'lobular-like' and hence more suited to colonisation of the leptomeningeal serosal membrane niche. In support of BCLM cells displaying a 'lobular-like' biology, the in vivo models of tumour dissemination via haematogenous route, showed BCLM cells had grew readily in the ovaries, and pituitary and adrenal glands, at higher rates than in the patients these derived from. This suggests that the biological adaptation which BCLM tumour cells undergo in the leptomeningeal space, confer an advantage to growth in endocrine organs[49], reflecting their shift to a more 'lobular-like' phenotype through the adherens junction and cytoskeletal defects.

Advances in the treatment of BCLM and leptomeningeal metastasis from other solid tumours are desperately needed. The standard-of-care BCLM intrathecal agent, methotrexate, was not particularly effective in BCLM PDOs treated in vitro, despite being used at concentrations similar to that achieved by intra-CSF injection. Although originally investigated in the adjuvant setting, methotrexate is not a commonly used chemotherapeutic agent in

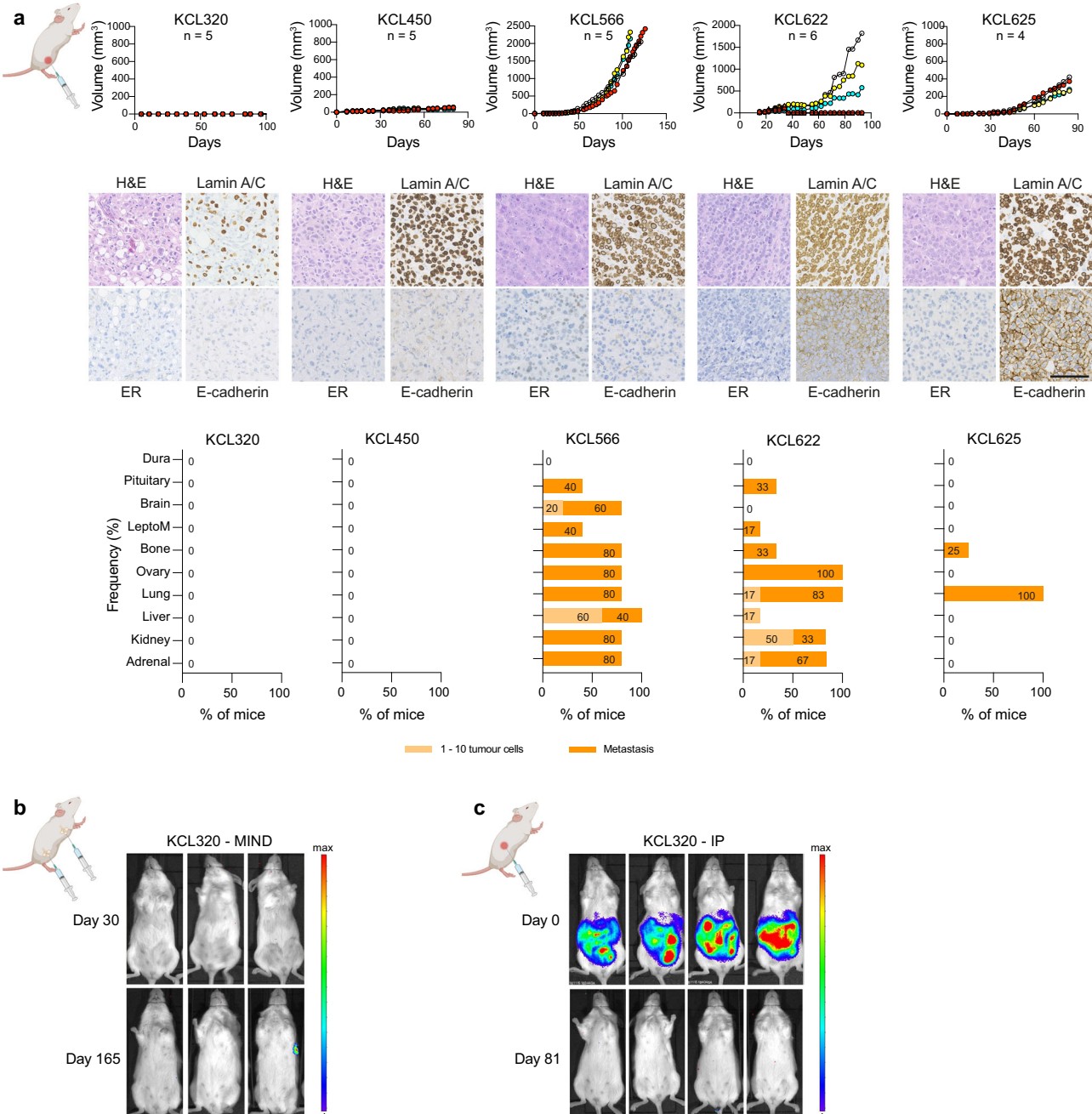

**Fig. 7 | Variable engraftment rate of BCLM PDOs in the orthotopic site.** Female NSG mice were inoculated with mChLuc2-labelled PDOs as follows. **a** Non-dissociated BCLM PDOs (~1 × 10⁶ cells) were injected transcutaneously into the 4th mammary fat pad (unilateral), $n = 4$–6 mice per PDO. Upper panel, tumour size was measured bi-weekly. Middle panel, mammary fat pad tumours stained for H&E, lamin A/C, ER and E-cadherin. Scale bar, 100 μm. Lower panel, bar charts show percentage of mice with metastatic dissemination in individual organs. Pale bars indicate % tissues with single or small cell clusters (1–10 cells), dark bars indicate tissues with metastatic deposits (>10 cells), source data are provided as a Source Data file. **b** 4 × 10⁴ dissociated KCL320 PDO cells per gland were inoculated via the mammary gland intraductal (MIND) route into the 3rd, 4th and 5th mammary glands, bilaterally (6 glands/mouse; $n = 6$ mice). Mice were IVIS imaged on day 0 and day 165 (endpoint). **c** 2 × 10⁶ dissociated KCL320 PDO cells injected intraperitoneally ($n = 4$ mice) with IVIS imaging at day 0 and day 81 (endpoint). Created with BioRender.com.

breast cancer, and the only randomised trial of intrathecal methotrexate in BCLM, failed to show a significant improvement in survival using this treatment[47]. Given the pressing requirement to discover novel approaches to treat BCLM, we describe the potential of CSF sampling and subsequent PDO derivation to enable discovery and functional validation of actionable targets. The genomic findings also highlight the potential clinical value of CSF cfDNA profiling for patients affected by this condition, and support the notion that BCLM may be better treated by discovery of blood-brain/

blood-CSF-barrier penetrant therapies rather than intrathecal cytotoxic chemotherapy. Importantly, the identification of cell-cell adhesion and cytoskeleton alterations provides insight to the biology of BCLM and its ability to flourish in this metastatic site, but also presents a potential 'Achilles-heel' for therapeutic targeting. The methodologies and models described here serve as a timely resource for the ongoing research into leptomeningeal metastasis with the overarching aim to improve therapeutic options and patient survival.

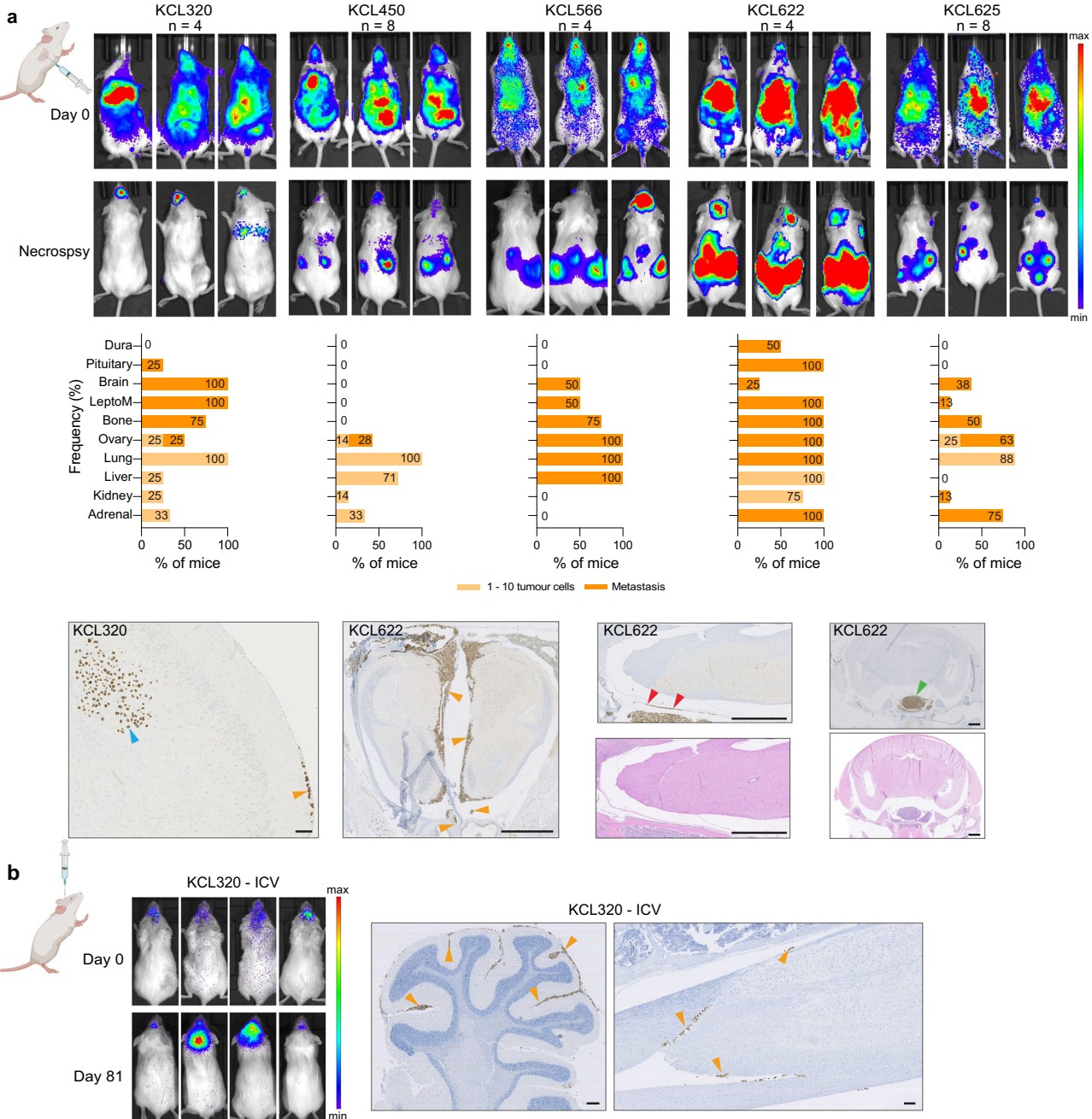

**Fig. 8 | Modelling BCLM by intracardiac or intracerebroventricular injection of PDOs.** BCLM PDOs were dissociated into single cells. **a** $0.5–1.0 \times 10^6$ cells inoculated into the left cardiac ventricle under ultrasound guidance. Upper panel, IVIS imaging immediately after cell injection (day 0) and at endpoint for each mouse (development of significant morbidity), day 40–210. Middle panel, metastatic site involvement assessed by lamin A/C staining of FFPE sections. Bar charts show percentage of mice with disseminated tumour cells in individual organs, with pale bars indicating with single or small clusters of cells (1–10 cells) and dark bars indicating tissues with metastatic deposits (>10 cells), source data are provided as a Source Data file. Bottom panel. IHC staining for human lamin A/C of whole cranium and spine FFPE sections showing tumour cells growing along the meninges (orange arrowheads) or in the brain parenchyma (blue arrowheads), in the dura (red arrowheads) and pituitary (green arrowhead). Scale bars; Left, 100 μm; other panels, 1 mm. **b** $1 \times 10^5$ cells inoculated intracerebroventricularly (ICV) (*n* = 4 mice). Left panel, IVIS imaging on day 0 and day 81 (endpoint) showing development of metastasis in two mice. Right panels, cranium and spine FFPE sections stained for human lamin A/C. Orange arrowheads indicate leptomeningeal involvement. Scale bars; 250 μm. Created with BioRender.com.

## Methods

### Patients and clinical sample collection
CSF and blood were concurrently collected from breast cancer patients at initial evaluation of BCLM (*n* = 21) as previously described[21]. Archived primary tumour and any metastatic recurrence tissues were retrieved. Written informed consent was obtained from all individuals in accordance with the Declaration of Helsinki under the following research ethics committee approved studies (REC ID 13/LO/1248, South East London Cancer Research Network, UK, and REC ID 14/LO/0292, Royal Marsden NHS Foundation Trust, UK). Histological subtype of primary tumours and BCLM PDOs were classified by independent breast pathologist review (I.R.).

### Sample handling and processing
CSF samples were collected in standard universal containers and stored transported on wet ice to the laboratory, for centrifugation

within 1 h of collection at 300 $g$ for 10 min. CSF supernatant was stored at −80 °C. The resultant CSF cell pellet was used for immediate entry to organoid culture (see below) or cells were re-suspended in 100 μL of 20% DMSO/FBS plus 100 μL OcellO primary organoid media (see below), and frozen to −80 °C at a cooling rate of −1 °C per minute using a Nalgene Cryogenic Freezing container. Extraction of CSF and plasma cfDNA, extraction of DNA from germline (buffy layer) and archival tissues, and ulpWGS for tumour purity assessment of cfDNA, has been previously described[21]. DNA from BCLM PDOs was extracted using Qiagen Blood and Tissue Kit.

### Whole exome sequencing (WES)
Germline, tumour tissue and organoid DNA were fragmented to 200 base pair (bp) length using a Covaris E Series instrument (Covaris, MA, USA), while cfDNA was not further fragmented. Sequencing libraries were made using SureSelect XT Low Input Library Preparation kit and Agilent Bravo Automated Liquid Handling Platform. SureSelect XT Low Input Dual Index P5 Indexed Adaptors were used for cfDNA to minimise index hopping[60]. Number of pre-capture PCR cycles was 8–14 based on input DNA quantity and quality (assessed by Agilent TapeStation) as advised by the manufacturers' protocol and run on an AB Veriti 96-well Thermocycler (Applied Biosystems, CA, USA). The target amount for hybridisation was 1000 ng. Exome capture was performed using the SureSelect XT Low Input Target Enrichment kit, Dynabeads MyOne Streptavidin T1 magnetic beads and the SureSelect XT Low Input Human All Exome V6 panel (60 Mb). The resultant libraries were run on the NovaSeq 6000 with an S2 flowcell (Illumina, San Diego, CA, USA) acquiring 2 × 100 bp paired-end reads. WES analysis was run through a bespoke pipeline using Nextflow v20.10.0[61]. Further analysis and plotting were performed in R 3.6.0.

**Alignment, duplicate removal and quality control.** Reads from exome sequencing FASTQ files were first trimmed with Trim Galore v0.6.6 (https://github.com/FelixKrueger/TrimGalore), a wrapper script for Cutadapt (https://doi.org/10.14806/ej.17.1.200) and FastQC (https://qubeshub.org/resources/fastqc). Reads were then aligned against the human assembly build GRCh38 using Burrows-Wheeler Aligner (BWA) mem v0.7.17, run with default settings. Duplicate reads were removed with Picard Tools v2.23.8 according to Genome Analysis Toolkit (GATK) Best Practices recommendations[62]. GATK v4.1.9.0[63] was used for base score recalibration. Sample coverage statistics were calculated with the CollectHsMetrics command in Picard. Coverage is displayed in Supplementary Fig. 1c and Supplementary Table 2. NGSCheckMate (downloaded March 15th 2018, commit number 1079908) was run to check for concordance between tumour and normal pairs for all patients, thus verifying sample identity[64].

**Variant calling.** A consensus variant calling method was used that included four variant callers: MuTect2[65], MuSE[66], LoFreq[67], and Strelka[68].

**MuTect2.** Before variant calling, a panel of normals was created by running MuTect2 (embedded in GATK v4.1.9.0) in unpaired mode on all germline samples in the study and aggregating the results with the CreateSomaticPanelOfNormals command. Paired tumour/germline variant calling was performed with MuTect2, which was provided with both the panel of normals and the Genome Aggregation Database (gnomAD) v3.1 for filtering. Resulting variants were further filtered on contamination fractions and orientation biases with the FilterMutect-Calls command. Only biallelic calls were retained for further processing. Variants were annotated with ANNOVAR (2016-02-01 version)[69]. PolyPhen-2[70] and SIFT[71] algorithms, embedded in ANNOVAR, were used for predicting the effects of non-synonymous variants on protein function.

**MuSE.** MuSE was run in exome mode on the tumour-germline pairs. The gnomAD database was provided for removal of common variants.

**LoFreq.** Tumour and germline BAM files were first run through LoFreq's indelqual command to add the required information for calling indels. Then the somatic command was used to perform the variant calling. The gnomAD database was once again used as a resource of common variants.

**Manta and Strelka.** Manta and Strelka were run, in that order, to perform indel calling and SNV calling, respectively. Both were restricted to the exome. Indel and SNV data were combined at the end of the run.

The resulting variants were merged and restricted to only those reported by two or more callers. Those variants were further filtered; (a) by limiting them to exonic or splicing regions and (b) by applying a set of quality filters as follows: a minimum of 5 supporting reads in the tumour sample, a maximum of 2 reads in the germline sample, a minimum coverage depth of 10 reads, and a minimum VAF threshold (as defined below). The two exceptions to this were low coverage (median coverage ≤50x) samples (CSF samples RMH010, RMH011 and KCL680) that had reduced read thresholds of 3, 1, and 7, respectively, and LoFreq variants for which the number of supporting reads in the normal sample is not reported. (c) A higher threshold was applied in each purity stratum to C > T and G > A variants to remove FFPE artefacts and (d), in cases where a variant passed these filters in one sample of a patient but not in another, the non-passing variant call was exempted from the filtering due to the additional evidence. Tumour mutational burden (TMB) was calculated for each sample by dividing the total number of these variants by the total megabases covered by the exome target panel.

Tumour purity and ploidy estimates were obtained from PureCN v1.16.0[72] except in the following situations (a) cfDNA samples, which had also undergone ulpWGS, where difference between PureCN and ulpWGS purity estimate > ± 0.15, (b) samples with PureCN ploidy > 3.5 or < 1.5, (c) tumour tissue DNA with purity estimate ≤ 0.35. For these samples, manual curation of purity and ploidy was performed to reach a consensus purity and ploidy estimate, by examining the ichorCNA and CNVkit copy number profiles, and the median VAF of somatic variants present in the sample.

High purity samples (purity > 0.5) were subjected to the highest filtering thresholds of 0.05 minimum VAF for non-FFPE samples and 0.07 minimum VAF for C > T and G > A SNVs in FFPE samples. For low purity samples (purity ≤ 0.2), the minimum VAF filters were adjusted to 0.01 (non-FFPE) and 0.03 (FFPE). For samples with purities between 0.2 and 0.5, intermediate VAF thresholds were set to 0.03 (non-FFPE) and 0.05 (FFPE).

**Mutational signatures analysis.** Consensus SNV calls from each sample were converted back into VCF format using the MutationalPatterns R package[73]. Cosine similarity and signature contribution were calculated based on the COSMIC SBS signatures v3.2[74].

**Cancer driver gene lists (used in Fig. 3 and Supplementary Fig. 14).** Cancer Gene Census (CGC) list[75] was downloaded from COSMIC database (https://cancer.sanger.ac.uk/census) on 4th July 2018 (719 genes). Breast cancer driver gene list (89 genes) was created by combining breast cancer drivers identified in two published studies[76,77]. OncoKB potentially actionable variants (Therapeutic Level 1–4) were identified by accessing https://www.oncokb.org[78] on 17th February 2022.

**Copy number alteration calling.** Copy number log$_2$ ratios were obtained from CNVkit v0.9.9[79], with each tumour sample run against a normal reference built from all matched germline samples in the study.

The segmented ratios from each sample were then adjusted for tumour purity and ploidy, which were determined as described above. A copy number ratio was assigned to each gene using CNVkit's gene-metrics function. To call copy number states, $log_2$ ratios were first standardised (normalised to standard deviation) within each sample and copy number states were called for each sample as follows:

Copy number alterations (CNAs) were defined using study specific $log_2$ ratio (standardised $log_2$ ratio (z)) thresholds as follows: amplification > 2.0; gain > 1.0 to ≤ 2; deletion < −2.5; loss <−1.5 to ≤ 2.5; neutral copy number state ≥−1.5 to ≤ 1.0.

For CNA proportion test, differences in CNA frequency between sample types were analysed with two-sided tests of equal proportions in R. Separately, GATK v4.1.9.0 haplotype caller was run on both tumour and normal samples to obtain B-allele frequencies. ASCAT v2.5.1[80] was provided with the same purity and ploidy estimates used with CNVkit and used to obtain allele-specific copy number calls for use in PyClone (see below).

**Genomic alterations in BCLM *vs*. metastatic breast cancer (MBC) cohort.** Mutation and copy number data from a cohort of 216 MBC samples which had undergone WES biopsy in the context of the SAFIR01 and SAFIR02 prospective trials[26] (https://www.cbioportal.org/study/summary?id=brca_igr_2015; downloaded 9th August 2022). Rate of genomic alteration (including non-silent mutation, amplification and deletion) in each cohort (BCLM *vs*. MBC) were compared by proportion Chi-square test, using R.

**Gene set enrichment analysis.** Genes enriched for alterations (including mutation, amplification and deletion) *vs*. matched primary tumour, 257 genes in CSF (altered in ≥ 4/21 of CSF samples and ≤2/21 matched archival tissue samples) and 160 genes in plasma (altered in ≥ 3/11 plasma samples and ≤2/21 matched archival tissues) were used as input lists for statistical overrepresentation test using the GO Biological Process (complete) annotation set of PANTHER database (http://www.pantherdb.org, analysis performed 12th July 2022). Enriched gene sets were retained if -$log_{10}$ p-value (Fisher's exact test) was ≥ 3.0 in either CSF or plasma list, and if the reference list (size of GO family) was ≥ 10.

**Subclonal reconstruction and clonal evolution modelling.** Subclonal reconstruction was performed with PyClone v0.13.1[81], clustering all filtered non-silent variants based on their allele frequencies, with adjustment for allele-specific copy number states (from ASCAT) and sample tumour purity estimates as described above. Variant clusters with corresponding cellular prevalence identified by PyClone were used as input to ClonEvol (Version 0.99.11)[82] run in R for clonal ordering and phylogenetic tree construction. Unless otherwise stated below, analysis parameters were: number of mutations per cluster ≥ 10; cluster VAF of > 0.05; monoclonal cancer initiation model; bootstrap, non-parametric subclonal test with a minimum probability that cluster is non-negative with a *p*-value set at 0.05. Adjustments to parameters for individual models were: number of mutations per cluster ≥ 2 (KCL650), ≥ 5 (KCL523), ≥ 19 (KCL499), ≥ 30 (KCL566); and cluster VAF > 0.01 (KCL499) and > 0.02 (KCL622).

**RNAseq analysis of primary tumour and matched BCLM PDOs**
Total RNA from archival primary tumours (FFPE) was extracted by AllPrep DNA/RNA FFPE Kit following macrodissection of involved tumour areas. Cell lines and early passage (passage number 3–5) PDOs (8–16 days post seeding), were harvested in RLT buffer and total RNA extracted using the RNeasy kit (QIAGEN) according to the manufacturer's instructions. RNA samples were quantified using Qubit 2.0 Fluorometer and RNA integrity was checked with Agilent TapeStation. RNA from PDOs underwent library preparation using NEBNext Ultra RNA Library Prep Kit for Illumina and sequenced on Illumina HiSeq

acquiring paired end reads. RNA from primary tumours underwent library preparation using NEBNext Ultra II Directional RNA Library Prep Kit for Illumina and sequenced on an Illumina NovaSeq 6000 S2 platform acquiring paired end reads. To evaluate the quality of paired-end RNA-sequencing data, FastQC and FastQ Screen[83] were run on fastq files and a summarised report was generated using MultiQC (v1.9)[84]. Reads were trimmed using Trim Galore (v0.6.6). Trimmed reads were aligned to the human reference genome GRCh38, using STAR 2.7.6a[85] with −*quantMode GeneCounts* and −*twopassMode Basic* alignment settings. Annotation file used for feature quantification was downloaded from GENCODE (v22) in GTF file format.

**Cell lines**
DU4475, MDA-MB-134-VI, T47D and SUM44PE were Isacke laboratory stocks, MCF7 $^{CDH1+/+}$ and MCF7 $^{CDH1−/−}$ cell lines were obtained from Professor Chris Lord (Institute of Cancer Research, UK), derived as previously published[86]. Identity of all cell lines were confirmed by short tandem repeat testing (GenePrint 10 ID System; Promega) (Supplementary Table 7). All cell lines were negative for mycoplasma contamination on regular testing using MycoAlert detection kit (Lonza). Cells were cultured in RPMI plus 10% FBS in 2D on tissue culture plates or in 3D in non-adherent suspension plates in a 1:1 mix of Matrigel (growth factor-reduced, BD Biosciences - 354230) and RPMI plus 10% FBS. Cell lines used for RNAseq were cultured in 3D in non-adherent suspension plates in a 1:1 mix of growth factor-reduced Matrigel and OcellO primary organoid media (see below). Total RNA was extracted as above.

**Conditioned medium from primary meningeal cells**
Primary human meningeal cells (ScienCell) were cultured in 2D monolayers on poly-L-lysine (ScienCell) coated flasks, using meningeal cell culture media (1401 and 1452, ScienCell). Once the cells reached 70–80% confluency, the cells were cultured in OcellO primary organoid medium for 24 h after which the conditioned medium was collected, centrifuged and passed through a 0.2 μm pore syringe filter and stored at −20 °C.

**Generation of BCLM patient-derived organoids (PDOs)**
The freshly collected CSF sample was transferred to sterilised 2 mL V bottom tubes and centrifuged at 300 *g* for 10 min. The cell pellet was resuspended in OcellO primary organoid medium to 10 μL. The composition of the OcellO primary organoid medium is as follows: Advanced DMEM/F12, 10 mM HEPES, 1× Glutamax, 1× B-27 without retinoic acid, 1× N-2 supplement, 20 ng/ml fibroblast growth factor 2 (FGF2), 50 ng/ml EGF (epidermal growth factor), 10 μM Y-27632, 1.25 mM N-acetylcysteine, 5 μM A83-01, 1x penicillin and streptomycin. An equivalent volume of Matrigel was added and mixed by pipetting. 10 μL of sample mixture was pipetted per well of 96-well round bottom suspension culture plate (pre-warmed to 37 °C) and placed in a tissue culture incubator for 30 min, prior to adding 150 μL PDO culture medium. PDO culture medium was composed of 50% primary organoid medium (OcellO), 50% meningeal conditioned medium (see above) supplemented with 10 ng/mL neuregulin-1, 10 ng/mL EGF, and for ER+ breast cancers (from primary tumour histology), 17β-oestradiol ($10^{−11}$ M). PDO culture media was changed every 7 days. Typically, organoids developed within 7–21 days after initial seeding and were first passaged (1:2) by dissociation using TrypLE Express Enzyme (ThermoFisher) and resuspension in OcellO primary organoid medium. From then on organoids were cultured in OcellO primary organoid medium supplemented with oestradiol, neuregulin-1 and EGF (without meningeal conditioned medium). Once cultures were expanded, there were re-passaged at 1:3–1:6 every 7 days. All established organoids were regularly tested for mycoplasma, and identity confirmed by short tandem repeat (STR) testing using GenePrint 10 (Promega) (Supplementary Table 7), with comparison to patient

germline or primary tumour DNA. For lentiviral transductions, organoids were dissociated into single cells using TrypLE, resuspended in OcellO primary organoid medium with the addition of lentiviral particles containing the PGK-H2BmCherry-IRES-Luc2 construct and polybrene (8 μg/mL). Cell/lentivirus mixtures were centrifuged at 800 g at 32 °C for 30 min, prior to culturing as suspension cells in low adherence 6-well plates for 48 h. Cells were then collected and centrifuged to remove media containing virus and cultured in 50% Matrigel droplets. After 14 days cells were dissociated to single cells and FACS sorted using a FACSDiva (BD Biosciences) to select for mCherry-positive cells and returned to PDO culture.

### In vitro PDO drug assays

Using Hamilton cold stage dissociated, BCLM PDO (3750 cells/well) or DU4475 (1500 cells/well) were plated in 384-well clear flat bottom plates in 50% Matrigel:50% media droplets. 35 μL of OcellO (PDOs) or cell line (DU4475) media were added to each well. 24 h later, media containing methotrexate (Enzo, ALX-440-045) in 0.2% DMSO or 0.2% DMSO alone (vehicle alone) was added to the wells and replenished on day 7. Cell viability was measured by CellTiter-Glo after 14 days.

### RTqPCR

Quantitative PCR reactions were performed using Taqman Gene Expression Assay probe (*B2M*: Hs00187842_m1; *CDH1*: Hs01023895_m1). The QuantStudio6-Flex sequence detection system was used to perform relative quantification, with all reactions performed in triplicate. Data analysis was performed using Applied Biosystems Design & Analysis software 2.6.0. *B2M* was used as endogenous control.

### Preparation of PDOs for histological analysis

Organoids growing in Matrigel droplets were prepared for histology by removing media and adding 4% paraformaldehyde solution to each well to cover droplets and placed at room temperature for 60 min. Organoids were then pipetted into a 15 mL Falcon tube, washed twice with PBS, collected by gentle centrifugation (150 g for 5 min), resuspended in 50–100 μL of warmed HistoGel (Thermo Fisher Scientific) and placed on Parafilm laid over ice for 5 min until solidified. HistoGel droplets containing organoids were then paraffin-embedded for sectioning and haematoxylin and eosin (H&E) and/or immunohistochemical staining (Supplementary Table 8).

### Establishment of patient-derived BCLM in vivo models

All animal work was carried out under UK Home Office Project Licenses 70/7413, P6AB1448A and PP4856884 granted under the Animals (Scientific Procedures) Act 1986 (Establishment Licence, X702B0E74 70/2902) and was approved by the "Animal Welfare and Ethical Review Body" at The Institute of Cancer Research (ICR). All mice were housed in individually ventilated cages and kept at 21 °C ± 2 °C, humidity level between 45 and 65% and light/dark cycle of 12 h. Mice were monitored daily by ICR Biological Services Unit staff and had food and water *ad libitum*. Mice were weighed at least two times per week. All experiment were performed with female 6–8 week NSG mice (Charles River). For PDOs KCL320, KCL450 and KCL566 sustained-release 17β-oestradiol pellets (0.36 mg/pellet, 90 day release, Innovative Research of America) were placed subcutaneously 7 days before PDO inoculation.

**Mammary fat pad inoculation.** Mice were anaesthetized using isoflurane, and placed supine on a heat pad. Forceps were used to gently lift the 4th mammary fat pad nipple, and undissociated PDOs in 100 μL total (50% PBS: 50% Matrigel) were injected into the fat pad underlying nipple. Tumour growth was measured twice weekly up to a maximum diameter of 18 mm. Tumour volume was calculated using the following formula: Volume = 0.5236 x [(width + length)/2]$^3$. Maximal tumour size/burden was not exceeded.

**Intraductal inoculation.** Intraductal inoculation was performed as previously described[87]. In brief, dissociated PDO cells were inoculated bilaterally into the 3rd, 4th, and 5th mammary glands (6 glands/mouse). Tumour growth was measured twice weekly up to a maximum diameter of 15 mm if a single tumour was present and 12 mm when multiple tumours formed. Maximal tumour size/burden was not exceeded.

**Intraperitoneal inoculation.** $2 \times 10^6$ dissociated PDO cells in 100 μL PBS were injected intraperitoneally.

**Intracardiac inoculation.** Fur was removed from chest prior to procedure. Mice were anaesthetized with isoflurane and secured onto a VisualSonics Vevo-770 ultrasound imaging platform with continuous electrocardiogram monitoring. Ultrasound gel was applied over the chest. Dissociated PDO cells were passed through a 40 μm sieve. $5 \times 10^5$ cells in 100 μL sterile PBS were injected into the left ventricle under continuous ultrasound guidance. To confirm successful inoculation IVIS imaging was performed within 30 min of injection.

**Intracerebroventricular (ICV) inoculation.** Mice were anaesthetised with isoflurane, prior to depilation and midline incision of skin overlaying the skull and injected with $1 \times 10^4$ dissociated PDO cells in 5 μL PBS into the right lateral cerebral ventricle at a rate of 2.5 μL tumour cells/min using a stereotaxic frame with pre-defined co-ordinates relative to bregma (x = 0.7 mm, y = + 0.6 mm, z = + 2.0 mm). Following injection, the skin incision was closed.

### Fixation, staining and imaging of tissues

Organs removed at necroscopy were fixed for 24 h in 10% neutral buffered formalin at room temperature for 24 h and paraffin-embedded for sectioning. For bone histology, soft tissue was stripped off prior to fixation in 10% neutral buffered formalin for 72 h at 4 °C and decalcified in Hilleman and Lee EDTA solution (5.5% EDTA in 10% neutral buffered formalin) for 7–10 days. Once decalcification was complete, midline sagittal sections (whole head and spine) or coronal sections (whole head) were cut using a scalpel, and tissues were paraffin-embedded for sectioning. Metastases were quantified from sections stained for human lamin A/C. Identification of dural and pituitary metastases was confirmed by pathology review (I.R.).

### Statistics

All statistical tests were performed in GraphPad Prism 9 or R statistical environment (v3.6.0). Error bars indicate ± standard deviation (SD). All comparisons between two groups were unpaired, and data was tested for normality by Shapiro-Wilk test. All *p*-values are two sided. Correlation tests were performed in R using the test for association between paired samples, using Pearson's product moment correlation coefficient and two-sided *p*-values. Unless otherwise stated the proportions testing was carried out in R using Chi-squared test. All measurements shown are from distinct samples.

### Reporting summary

Further information on research design is available in the Nature Portfolio Reporting Summary linked to this article.

## Data availability

The RNAseq data is deposited in Sequence Read Archive (SRA) under open access - ID number, PRJNA988939. The processed data from whole exome sequencing are provided in Supplementary Data files 1–5, including all somatic variants and copy number alterations identified. All other data generated in this study are provided in the Supplementary Information, Supplementary Data and Source Data files which accompany this manuscript. We do not have permission from the patients to publicly deposit the raw exome sequencing data.

Correspondence and requests for further data or materials should be addressed to Professor Clare Isacke (clare.isacke@icr.ac.uk). Requests for raw exome sequencing data should be made through formal data access request describing the nature of the proposed research and extent of data requirements. This will then be reviewed by the trial management group and study sponsors. If approved, recipients are required to enter a formal data sharing agreement, which describes the conditions for data release and requirements for data transfer, storage, archiving, publication, and intellectual property. Graphical depictions were created with BioRender.com.

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

## Acknowledgements

This study was supported by: Medical Research Council Clinical
Research Training Fellowship - MR/P001564/1 (A.F., C.I.), NIHR Biome-
dical Research Centre (BRC) at the Royal Marsden and the ICR post-
doctoral support funding - W94500 (A.F.), Breast Cancer Now Pro-
gramme Funding to the Breast Cancer Now Toby Robins Research

Centre - CTR-Q4-Y1-5, CTR-Q5-Y1 (C.I., N.T., A.N.J.T.) and to the Breast Cancer Now Research Unit at Kings College London - KCL-Q2-Y1-5, KCL-Q3-Y1 (A.N.J.T.), Cancer Research UK Centre Grant funding to the Kings College London - CRUK RE15196-9 (A.N.J.T.). We would like to thank Naomi Guppy and her team in the Breast Cancer Now Toby Robins Research Centre Nina Barough Pathology Core Faculty for histopathology support and the following facilities at the ICR; Tumour Profiling Unit, Biological Services Unit, FACS and Light Microscopy Facility. We thank Eleanor Knight, Daniela Nova and Rebecca Marlow in the Breast Cancer Now Toby Robins Research Centre Patient Derived Models facility for their help in establishing the PDO culture methodologies and OcellO for providing the PDO primary organoid medium. We thank John Alexander, Erle Holgersen and Hui Xiao in Syed Haider's team for their contributions to data analysis. We thank Phil Bland (ICR) for mouse intraductal injections. For patient enrolment and sample collection at Kings College London and Guys and St Thomas's NHS Foundation Trust, we thank clinicians S. Irshad, H. Kristeleit, J. Mansi, M. Nathan, E. Sawyer, A. Swampillai, M. Harries and the study nurses and staff, and R. Liccardo, R. Marlow, and V. Shah for patient enrolment and sample preparation collection. For patient enrolment and sample collection in Royal Marsden NHS Foundation Trust, we thank clinicians S. Johnston, M. Parton, M. Robert, L. Ulrich and A. Okines and the study nurses and staff, A. Bambra, D. Kelly, S. Thompson.

## Author contributions

Conceptualization A.F., A.N.J.T., C.M.I.; Supervision S.H., A.N.J.T., C.M.I.; Funding acquisition A.F., A.N.J.T., C.M.I.; Investigation A.F., M.I., A.M., D.V., T.A., I.R., N.C.T., S.H., A.N.J.T., C.M.I.; Writing - original draft A.F., M.I., A.N.J.T., C.M.I.; Writing - review and editing A.F., M.I., A.M., D.V., T.A., I.R., N.C.T., S.H., A.N.J.T., C.M.I.

## Competing interests

The authors declare no competing interests.
