## [Peer review file · Nature Communications]

REVIEWER COMMENTS

Reviewer #1 (Remarks to the Author): expertise in breast cancer genomics

The authors analyzed WES data from matched CSF (n=21), primary tumor (n=18), extracranial metastases (n=8), plasma cfDNA (n=11) from in total 21 patients in order to study the heterogeneity between leptomeningeal and extracranial tumor sites in breast cancer. Their analyses point to an association between a lobular-like breast cancer phenotype and leptomeningeal metastasis. Finally, they established patient-derived organoid models for 5 patients to identify therapeutic approaches targeted to this particular type metastasis. This is an interesting paper. However, there are several methodological considerations that needs to be adressed relating to the genomic analysis as seen below:

1. In relation to Figure 1c,
How was the mutational signature analysis for SNVs performed?

How many mutations were used as input? Given that this is only WES data, it is very likely that there is not enough SNVs to perform this analysis. Merging data from all patients is also misleading.

Did all these patients received the same type and amount of chemotherapy? Mutational profiles in trinucleotide context (96-channel) should be provided.

2. In relation to Figure 1e,
PCDH9, PCDH17 and KLHL1 are all on 13q21. Could this just be a loss of 13q enriched in CSF when compared to primary?

Similarly, could this be a loss of 16q in CSF when compared to plasma sample rather than just CDH1 loss?

Was there are any focal losses or amplifications on these genes?

How were these genes identified?

3. In relation to the ERBB2 amplification being shared by CSF and primary, the authors should acknowledge the limitation of their data. WES does not provide enough resolution to compare the structural variants that create the HER2 amplicon.

4. How was the copy number profiles of matched CSF, plasma and primary tumors compared? Copy number profiles of all CSF, plasma and primary tumors for each patients should be provided as supplementary material.

5. Are the phylogenetic trees based on SNV/Indels? Or do they include CNAs as well?

5. How was homozygous deletions identified? Cnvkit does not give allele-specific copy number.

6. In relation to Figure 2, only CSF-specific or shared mutations are indicated in the bar plot. Was there no mutation/alteration specific to primary/plasma sample?
Does the reference metastatic breast cancer dataset match the current cohort of 21 patients in terms of the specific breast cancer subtypes? 11/21 patients in this cohort is lobular which is enriched for CDH1 alterations. The enrichment of CDH1 compared to the reference dataset might be subtype-specific.

7. Almost all the CTNNA1 alterations seems to losses. Only one is a nonsense mutation according to

Figure 2. Are these losses focal?

8. The authors say "CTNNA1 and CDH1 aberrations displayed mutually exclusivity in BCLM samples". Is this statistically significant?

9. 8p loss is fairly common in breast cancer? From what is presented, it is difficult to see that ARHGEF10 or DMTN are the genes of interest in 8p? How focal are these events?

10. A lot of missense mutations are reported in the section titled "Biological pathway analysis shows enrichment for cytoskeletal aberrations in BCLM". For example, for DMTN, SPTA1. How do we know if these mutations are deleterious or of any functional relevance?

11. In section "BCLM subclones seed early during primary tumour evolution", the authors say their analyses revealed "(a) rather than a late-metastatic seeding event, BCLM disseminates early from the primary tumour, (b) CSF and plasma cfDNA display divergent evolution, often sharing branches with other metastatic sites but possessing distinct mutational repertoires, ". How many cases of 21 patients fall into these? These should clearly be indicated in text and in Figure 3c and S. Figure 4. The way they are the figures are very difficult to follow.

12. The authors say "Furthermore, two CSF samples, KCL566 and KCL148, exhibited high TMB (>10 mutations/Mb) (Fig. 2) predictive of response to immune checkpoint inhibition. It was noted that both these TMB-high cases had truncal MLH1 aberration however substantial elevation of TMB was seen in CSF cfDNA compared to matched tissues indicating further accumulation of mutational events in the leptomeninges". Could the increase in TMB in the CSF be because multiple distinct clones with a high TMB are found and detected in the CSF?

13. In section "Development and characterization of BCLM PDOs" the authors say "The high concordance between CSF cfDNA and CSF derived tumour cell DNA, confirms that CSF cfDNA represents a valid surrogate for the rarely available BCLM genetic and biological material.". Compared to what? What is the overlap with other samples available for these 5 patients.

14. The authors note lower expression of CDH1 and CTNNA1 in PDOs. How were these genes identified? Is this significant when differential expressed genes are identified in an unbiased manner?

Reviewer #2 (Remarks to the Author): expertise in bioinformatics analysis of genomic evolution

The manuscript by Fitzpatrick et al, describes a rare event in breast cancer, i.e. breast cancer leptomeningeal metastasis (BCLM). The authors study the inter-tumoral heterogeneity between BCLM (CSF cfDNA), extracranial metastases (plasma cfDNA or metastatic tissue) and the primary tumour, putting together a unique compendium of data from these sites. Furthermore, the authors manage to culture patient derived organoids (PDOs) from disseminated tumor cells in the CSF, which they could show have similar mutational spectra as those found in the cfDNA, which is remarkable.

Following WES the authors compare the copy number profiles between the PDOs and their matched CSF cfDNA samples and saw high level concordance and around 81.5% of the detected variants were present in both the sfDNA and the organoids. This is a spectacular result as it shows that CSF cfDNA represents a valid surrogate for the rarely available BCLM genetic and biological material. I am missing here a little comparison of the mutational spectre in the PDOs to that of the primary tumor. Loss of E-cadherin staining and reduced CDH1 expression was seen in one PDO by Immunohistochemical staining and transcriptional profiling of the PDOs and matched primary tumours showed specific differences as loss of membrane associated α -catenin staining and reduced CTNNA1 expression in other PDOs. However, it is known that PDOs differ substantially from primary tumors and much of these differences are due to culturing conditions. In this paper, the authors interpret the observed in the PDO qualities as directly characteristics of the DTCs, which is extrapolation.

Also notably the loss of ER expression has been observed multiple times when culturing primary tumors, and experimental conditions need to be carefully controlled. Can the authors show that when they culture ER positive tumors with these conditions, the ER status is preserved?

Minor comments:

I read easily the figures and the legends to figures, which I find clear and well illustrating the experimental flow. The text of the paper is however full of details that obscure the main message, and the comparisons made.

Some wording:

In "Consequently although 21/21 of CSF cfDNA samples had sufficient tumour content only 11/21 paired plasma cfDNA samples were suitable for WES.»

Replace «sufficient» with «detectible», not clear sufficient for what

Reviewer #3 (Remarks to the Author): expertise in breast cancer brain metastasis modelling

Fitzpatrick et al. describes genomic and pre-clinical modeling of breast cancer leptomeningeal metastases (BCLM), revealing that BCLM acquires a "lobular-like" phenotype. The authors have assembled a cohort of 21 patients with BCLM, from which cell free DNA (cfDNA) from CSF and plasma (subset of 11 patients) were available. Tissue from primary breast tumors and extracranial metastases (when available) was also analyzed. Whole exome sequencing (WES) was performed on the cfDNA samples (CSF and plasma), primary tumors and extracranial metastases. Importantly, the authors have established patient-derived organoids (PDOs) from cells purified from CSF and characterized these models for their ability to form leptomeningeal metastases in mice. The authors conclude that BCLMs exhibit an enrichment of specific mutations (e.g. CDH1, ARHGEF10, CTNNA1) relative to primary breast tumors. They also highlight the observations that many mutations observed in cfDNA from CSF implicate genes that encode constituents of cellular junctions, regulators of cytoskeletal architecture and modulators of cell migration/invasion.

The study of leptomeningeal metastases is an important area of research that requires concerted efforts to develop models to better understand the molecular underpinnings of these lesions. The authors should be congratulated for this informative and well written manuscript that advances our understanding of leptomeningeal metastases. However, there are several points that the authors can address to further strengthen the manuscript.

Main Points:

1) The authors use strong language throughout the manuscript such as, but not limited to "We found that BCLM samples, whether derived from ILC or non-lobular primary tumours, were enriched for alterations in CDH1". When examining the data presented in Figure 2 and Supplemental Figure 2, of the N=7 primary breast tumors included in the analyses that did not have CDH1 alterations at baseline, only 2 gained such an alteration in the specimen collected from CSF (1 loss, 1 stop-gain mutation). Furthermore, when looking at CTNNA1, of N=13 patients who had no CTNNA1 alterations in the primary tumor, CTNNA1 deletion was identified in 2 matched CSF specimens. In one primary tumor where CTNNA1 was lost, it was no longer lost in the CSF specimen. Ideally, a higher number of specimens should be collected to develop a statistically-based argument to substantiate these claims. If this is not feasible, the authors should qualify the statements made throughout the manuscript.

2) The OncoKB analysis where the authors claim that 57% of patients acquired therapeutic Level 1 alterations in the CSF specimens appears to be overstated. For example, while PIK3CA E542K and H1047R mutations are indeed described as Level 1 alterations, other PIK3CA mutations such as N1044K or E39K mutations are not listed in OncoKB's database. Other examples include ATM, for which mutations only have Level 1 evidence in prostate cancer, or BRCA2 mutations which are listed as Level 3A mutations in breast cancer. These analyses should be revised to accurately reflect the true

actionability of these newly identified mutations.

3) In Figure 4A, the authors claim that E-cadherin expression is lost by IHC (Fig. 4a) and RNASeq (Fig. 4c) analysis when comparing between the primary mammary tumor (positive) and the KCL566 PDO (negative). The RNAseq data does suggest a significant downregulation of CDH1 levels; however, the IHC image presented in the Figure appears to show some membranous E-cadherin staining. When injected into the mammary fat pad, the KCL566 PDO gives rise to E-cadherin negative mammary tumors (Fig. 5a). Can the authors comment on this apparent variability?

4) In the analyses presented in Figures 5 and 6, the authors describe differences between dural, pituitary, brain and leptomeningeal metastases. The authors are encouraged to include representative images demonstrating these various metastatic lesions from the 5 PDO models in the revised manuscript (from the cardiac-injected cohorts – Fig. 6A). Additional text providing more detailed methodology, particularly for the identification and determination of pituitary metastases, would be helpful. Are the pituitary metastases that develop following intra-cardiac injection of the PDO models similar to the skull-based lesions described from leptomeningeal metastasis models of medulloblastoma (Garzia et al. Cell, 2018)?

5) The authors provide a careful characterization the metastases that are formed following mammary fatpad and intra-cardiac injections for all 5 PDO models. However, only the KCL320 model was injected intracerebroventricularly. Do all 5 PDOs form only leptomeningeal metastases following this route of injection? For example, is there any difference in the ability to form LM using models that have no (KCL320, KCL450), low (KCL566) or high (KCL622 and KCL625) E-Cadherin levels? For the KCL320 model, what is the penetrance it of mice that develop LM when the cells are injected intracerebroventricularly (appears to be 50%: Fig. 6b).

6) An open question in the field of LMD is whether cells floating in the CSF are true proxies for the disease, or whether these cells are simply the ones not able to survive and proliferate in contact with the arachnoid/pia mater. Remsik et al. 2022 (Cancer Rep.) has published data beginning to solve this question, associating floating cells with aggressivity in the context of LMD. In relation to comment 5 above – do the 5 PDO models, when injected intracerebroventricularly, give rise to LM that shed cells into the CSF to varying degrees? The morphologies of the organoids are quite different, with some forming compact organoids (KCL320, KCL566, KCL625), slightly diffuse (KCL450) and very diffuse (KCL622) organoids. Would this relate to the extent of LMD in the various PDOs? It seems that these models would be very useful in this regard. It is interesting that the KCL622 model seems to generate the greatest degree of metastases following cardiac injection; whereas the KCL566 model produces more spontaneous metastases.

7) The authors provide intriguing data suggesting that ROS1 inhibition may be a useful therapeutic strategy for LMD with this lobular-like phenotype. This notion has important translational implications and would benefit by being further elucidated. Have the authors assessed expression of ROS1 in their PDO models? This would be interesting data to show and subsequently correlate sensitivity of the PDO models to ROS levels. The in vitro sensitivity of the PDO models is interesting; however, it would be important to assess responsiveness to ROS1 inhibition in vivo with several of these models. If possible, a model of LMD that has high and low/no expression of ROS1, but still derives from a lobular/CDH1-mutated breast cancer, would be worthwhile studying.

Reviewer #4 (Remarks to the Author): expertise in development of PDO models

In the manuscript presented for review, "Genomic profiling and pre-clinical modelling of breast cancer leptomeningeal metastasis (BCLM) reveals acquisition of a lobular-like phenotype," Fitzpatrick and colleagues report the whole exome sequencing of breast tumors in patients with leptomeningeal

metastases, along with sequencing of matched cell-free DNA from cerebrospinal fluid (CSF circulating tumor DNA) and serum (circulating tumor DNA), and DNA isolated from other metastatic lesions. The researchers compare the mutational profiles from each specimen type (e.g., tumor DNA from CSF vs. plasma) to explore the mutational spectrum and different evolutionary pressures, such as from chemotherapeutic treatment. This is a well-controlled study with added power from the analysis of the various tumor sites (primary, metastatic, and circulating DNA). Furthermore, the authors report development of primary ex vivo cell models (3D matrigel embedded) established from circulating CSF tumor cells that closely mimicked the phenotype of matched primary tumors. Furthermore, xenografts were successfully generated from the 3D organoids (injected into NSG mouse mammary fat pads), or dissociated cells into cardiac ventricles or cerebral ventricles that permitted analysis of growth, metastatic spread, and testing of chemotherapeutic agents. The authors correctly describe the grim prognosis of patients with leptomeningeal metastasis, and the findings and models they describe here are indeed an important resource for studying the biology of these rare tumors. The results contribute important information to the paucity of that surrounding this rare but highly lethal condition. This reviewer supports the publishing of this manuscript with only very minor revisions.

Suggestions:

On first read, I found the abstract a bit strong on lingo and it was a bit confusing to understand what the researchers had done and what this article contained. It needs minor refinement.

In the introduction, when describing lobular breast cancer, it would be helpful if the authors mentioned that the loss of E-cadherin is a general distinguishing characteristic of lobular breast cancers, as this is important for understanding the mutational profile of CDH1 (Fig 1e). The authors discuss this later, but it would be helpful here.

Authors need to define PDOs and give critical details of their generation in the narrative, as these can be generated in a variety of different ways.

What is Ocello primary cell medium? Is it proprietary? What are the key components that distinguish this medium?

Response to reviewers

Reviewer #1 (Remarks to the Author): expertise in breast cancer genomics

The authors analyzed WES data from matched CSF (n=21), primary tumor (n=18), extracranial metastases (n=8), plasma cfDNA (n=11) from in total 21 patients in order to study the heterogeneity between leptomeningeal and extracranial tumor sites in breast cancer. Their analyses point to an association between a lobular-like breast cancer phenotype and leptomeningeal metastasis. Finally, they established patient-derived organoid models for 5 patients to identify therapeutic approaches targeted to this particular type metastasis. This is an interesting paper. However, there are several methodological considerations that needs to be adressed relating to the genomic analysis as seen below:

1: In relation to **Figure 1c**, how was the mutational signature analysis for SNVs performed?

Response: As described in the Methods section 'Whole exome sequencing (WES)' subheading '(3) Mutational signatures analysis', single base substitution signatures were analysed using the MutationalPatterns R package. The input for the analysis were all consensus single nucleotide variants (SNV) passing filtering steps.

(a) How many mutations were used as input? Given that this is only WES data, it is very likely that there is not enough SNVs to perform this analysis. Merging data from all patients is also misleading.

Response: The reviewer raises an important point, that given only 1% of the whole genome is analysed during whole exome sequencing, this gives a limited view of mutational processes. However we note that mutational signature analysis using WES data is widely used and in particular, landmark analyses by the PCAWG consortium (Alexandrov et al., 2020) characterised mutational signatures from 23,829 samples, the majority of which (81%) were whole exome sequenced. The resultant COSMIC single base substitution (SBS) figures are based on these data.

Regarding the number of SNV available for mutational signature analysis, the SNV burden for the breast cancer samples in Alexandrov *et al* was 1-2 SNVs per megabase. Our analysis used the filtered SNVs, with a median number of SNVs of 112 per sample (CSF); 116 per sample (plasma); 76 per sample (primary tumour); and 70 per sample (metastatic tumour) (this data is shown in Supplementary Table 2). These equate to an SNV burden of 1.1 – 1.9 per megabase, therefore similar to the Alexandrov *et al* study.

For Fig. 1c, we merged the data from multiple patients to describe the chemotherapy related signatures found in the sample types. We appreciate that the decision to merge patients could introduce some bias, however we intend simply to provide an overview from a small number of patients, and appreciate it is difficult to draw firm conclusions. We have toned down the conclusion in the text on p.5 paragraph 2. Furthermore, a new table (**new Supplementary Table 4**) has been provided showing the total number of variants per sample used for mutational signature analysis, the chemotherapy-related signature contributions per sample, and mean of the total chemotherapy contributions across sample types (data which is shown in Fig 1c).

(b) Did all these patients received the same type and amount of chemotherapy? Mutational profiles in trinucleotide context (96-channel) should be provided.

Response: Following on from above, Supplementary Table 4 additionally shows the cytotoxic chemotherapy agents received by each patient. Most patients received anthracycline, cyclophosphamide and taxanes. In addition, a subset of patients received 5FU and a subset patients received platinum drugs, eribulin and/or vinorelbine.

As requested, we have provided the trinucleotide context (96-channel) profiles for all samples (**new Supplementary Fig. 2**).

2: In relation to **Figure 1e** -

(a) PCDH9, PCDH17 and KLHL1 are all on 13q21. Could this just be a loss of 13q enriched in CSF when compared to primary? Similarly, could this be a loss of 16q in CSF when compared to plasma sample rather than just CDH1 loss? Was there are any focal losses or amplifications on these genes?

Response: We thank the reviewer for highlighting this point. The volcano plots in Fig. 1e display the comparison of CNA frequency between sample types, on gene-by-gene basis. To clarify this analysis, we have now included a new figure (**new Supplementary Fig. 3**) showing the significant values by genomic position (equivalent to the p-value shown in Fig 1e).

The statistical analysis for the comparison of CNA frequency between sample types employed was a two-sided test of equal proportions, and genes are annotated on the Fig. 1e volcano plot if significance ($-\log_{10}$ p-value) was ≥ 1.5 (equivalent to p-value ≤ 0.0316). Where individual gene names are shown, p-value was significant for this gene and not neighbouring genes. Where cytoband locations are shown, a broader set of neighbouring genes were significantly enriched for CNA events, and genes of interest are contained in brackets.

In the revised manuscript, we have provided a new table (**new Supplementary Table 6**) which was submitted solely for reviewers during the 1st submission, showing the standardised log₂. This table shows the standardised log₂ ratio per gene for each sample. Using this data, the genomic breadth of CNA events can be inferred. It is correct that the gene losses noted on 16q were predominantly broad events, and often these events were arm level. It is our error that on Fig 1e the individual gene names are shown as instead of cytoband location in relation to 16q events, we thank the reviewers for raising this. Fig. 1e has now been corrected to show the cytoband locations in 16q with significant event enrichment, annotating genes of interest within these regions. In response to question regarding 13q losses, only the genes annotated (KLHL1, PCDH9, PCDH17) reached a $-\log_{10}$ p-value ≥ 1.5 for comparison of copy number event rate between CSF and plasma, therefore only these genes were annotated on the plot rather than cytoband locations. The size of the regions exhibiting 13q copy number loss in the relevant samples varied between 6 – 95 Mb, therefore not all were arm level.

We have revised the text on p.5 final paragraph to clarify these findings (see altered text in bold).

Compared to primary tumours, CSF cfDNA more frequently lost 13q regions containing PCDH9 and PCDH17 (members of the cadherin superfamily) and KLHL1, an actin binding protein with a role in cytoskeleton reorganisation. Broad 16q regions containing the classical cadherin CDH1 and other cadherins showed more frequent loss in CSF than plasma cfDNA samples.

(b) How were these genes identified?

Response: Please see response above and the clarification provided in the revised Fig. 1e legend

3: In relation to the ERBB2 amplification being shared by CSF and primary, the authors should acknowledge the limitation of their data. WES does not provide enough resolution to compare the structural variants that create the HER2 amplicon.

Response: We thank the reviewer for this important point. The text has now been clarified to acknowledge the limitation, on p.6 paragraph 1 as follows:

Similar to findings from brain metastasis cohorts (Brastianos et al., 2015), no new *ERBB2* genomic amplification events were found in CSF samples, **with the caveat that WES cannot identify complex structural variants or chromothripsis events leading to *ERBB2* amplification.**

4. How was the copy number profiles of matched CSF, plasma and primary tumors compared? Copy number profiles of all CSF, plasma and primary tumors for each patients should be provided as supplementary material.

Response: A challenge faced in comparing copy number states between sample types is the variability in tumour purity, particularly for liquid biopsy (CSF and plasma) samples. To overcome this we employed a sample-adjusted copy number calling approach, with log₂ ratios adjusted for sample purity and ploidy, to provide the most precise copy numbers calls. Thereafter, log₂ ratios were standardised (z) within each sample (normalised to standard deviation) and log₂ thresholds for calling copy number states were defined as described below. This has now been clarified in the Methods '(5) Copy number alteration calling'.

The following clarification for this methodology has been added to 'Methods, Whole Exome Sequencing, (5) Copy number alteration calling' section on p24 paragraph 3.

Copy number alterations (CNA) were defined using study specific log₂ ratio (standardised log₂ ratio (z)) thresholds as follows: amplification >2.0; gain >1.0 to ≤2; **deep** deletion <-2.5; loss <-1.5 to ≤2.5; neutral copy number state ≥-1.5 to ≤1.0.

The resultant copy number states were sense-checked against known against breast cancer reference genomes such as (Curtis et al., 2012) to ensure the thresholds chosen were biologically representative. We further refined the log₂ thresholds by sense checking with manual inspection of read counts using IGV for deep deletions. As a result, there was more stringent log₂ ratio threshold for deep deletions, which helps to mitigate for the absence of allele-specific copy number information from CNVkit.

A clarification for this methodology has been added to 'Methods, Whole Exome Sequencing, (5) Copy number alteration calling' section on p24 paragraph 3.

The resultant copy number states were sense-checked against known against breast cancer reference genomes such as (Curtis et al., 2012) to ensure the thresholds chosen were biologically representative. We further refined the log₂ thresholds by sense checking with manual inspection of read counts using IGV for selected amplicons.

5a: Are the phylogenetic trees based on SNV/Indels? Or do they include CNAs as well?

Response: Phylogenetic trees were generated using PyClone, which defines SNV/Indel variant clusters using mutant read counts combined with segmental and allele-specific copy-number data and tumour content estimates. PyClone groups somatic mutations into putative clonal clusters, and estimates their cellular prevalence within the sample sequenced. Although copy number data is key to refining the clonal cluster compositions, the evolution of CNA events is not able to be determined. This has approach has been clarified in the narrative on p.9 paragraph 3.

To investigate the **clonal** evolution of BCLM, **variant allele frequencies** and **allele-specific copy number states** were used to **infer mutational subclones** and **compose phylogenetic trees (Fig. 3c; Supplementary Fig. 4).**

5b: How was homozygous deletions identified? Cnvkit does not give allele-specific copy number.

Response: It is correct that due to the absence of allele-specific copy number from CNVkit, it is not possible to directly determine homozygous deletions. We have now corrected this in the text and defined these as 'deep deletion'.

As described in our previous response, we defined stringent study-specific log₂ ratio thresholds for defining copy number states, an approach used by large scale studies such as METABRIC in breast cancer (Curtis et al., 2012) which we believe is a more accurate method of determining true copy number states than using global thresholding on log₂ ratio thresholds, particularly when using samples of varying tumour purities.

6: In relation to **Figure 2**, only CSF-specific or shared mutations are indicated in the bar plot. (a) Was there no mutation/alteration specific to primary/plasma sample?

Response: Fig. 2 shows the cancer-associated gene mutations/CNAs most frequent in CSF samples, to demonstrate the cancer gene alterations most prevalent in the leptomeningeal metastatic site. An extended version of this plot is shown in Supplementary Fig. 4, which shows CSF side-by-side with paired plasma, primary tumour and metastasis samples for each case. We chose not to display the most frequent mutations/CNAs in plasma cfDNA or primary/metastatic tumours since we believe these are better presented by larger published cohorts of primary breast cancers such as TCGA or metastatic breast cancers cohorts such as (Yates et al., 2017). However for interest we have added a new table (**new Supplementary Table 7**) which displays the cancer-associated gene mutations/CNAs most frequent in plasma cfDNA samples, and the rate of which these were unique to plasma samples.

Further, looking with a broader view of all genes (not solely cancer associated genes) Fig. 3a shows the pathways/biological processes most altered (by mutation/CNA) in plasma samples, demonstrating that in contrast to CSF, the processes of 'UV response', 'mammary gland proliferation, protein deubiquitination proliferation', 'protein deubiquitination', and 'organonitrogen catabolic response', were enriched for genomic alteration in plasma. These were genomic changes additional to those detected in the matched primary tumours, therefore reflect the processes which have evolved at the metastatic sites.

(b) Does the reference metastatic breast cancer dataset match the current cohort of 21 patients in terms of the specific breast cancer subtypes? 11/21 patients in this cohort is lobular

which is enriched for CDH1 alterations. The enrichment of CDH1 compared to the reference dataset might be subtype-specific.

In response to the second part of the reviewer's question, histological subtype data was not available for the reference metastatic breast cancer dataset (MBC) cohort (Lefebvre et al., 2016), however all breast cancer histological types were recruited to the contributing studies. Receptor subtype data was however available for the MBC cohort and the dataset was overall similar to our cohort (ER+ HER2- 66% MBC and 62% BCLM; TNBC 24% and 24%, HER2+ 7% and 14%).

We anticipate that the percentage of lobular histology in MBC cohort was similar to the total breast cancer population i.e. 10-15%. Given that the BCLM cohort almost certainly has a higher proportion of patients with primary tumours of lobular histology (52.4% - see Supplementary Table 1), the reviewers is right to suggest this as an explanation for the *CDH1* mutation enrichment. We have acknowledged this in the statement on p6 final paragraph

Owing to the enrichment for lobular breast cancers in our BCLM cohort (Table 1, Supplementary Table 1) the high rate of *CDH1* (E-cadherin) mutations (52%) **was an expected finding**. Deleterious *CDH1* mutations are an early driver event in the majority of invasive lobular breast cancers (ILC), leading to defective adherens junctions (Ciriello et al., 2015; Desmedt et al. 2016) **The unexpected finding** was deleterious alterations of either *CDH1* or another key adherens junction component *CTNNA1* (α -catenin) in 55% of the non-lobular BCLM cases (Fig. 2).

However in addition to the expected *CDH1* mutations, we also noted some unexpected *CDH1* mutations occurring in non-lobular cancers, and in one case was BCLM-private, adding to the hypothesis that development of BCLM is enhanced in *CDH1*-defective cancers. Further to this, the finding of *CTNNA1* deleterious alterations, which is a suggested alternative mechanism to become 'lobular', adds to characteristics, supports our hypothesis that BCLM genomic events may lead to a lobular-like phenotype.

7: Almost all the *CTNNA1* alterations seems to losses. Only one is a nonsense mutation according to Figure 2. Are these losses focal?

Response: In addressing the reviewers question we refer to the copy number log₂ ratios from new table (**new Supplementary Table 6**) for the 3 cases which we have called as having *CTNNA1* deep deletions:

- KCL622 *CTNNA1* log₂ ratio of -3.99, neighbouring genes -1.52
- KCL625 *CTNNA1* log₂ ratio of -11.2 and neighbouring gene SIL1 , flanking genes -1.6
- KCL650 *CTNNA1* log₂ ratio of -2.97 and neighbouring genes -1.8

Based on our study specific log₂ ratio thresholds for copy number state as shown above, *CTNNA1* deletion events (log₂ ratio <-2.5) were focal, although arising within 5q segments with heterozygous loss (log₂ ratio <-1.5 to \leq 2.5;).

Although a small number of cases, we believe this finding is worth noting, in addition to the detection of a *CTNNA1* truncating mutation (p.S188*) in KCL680 with high allele fraction in CSF (46.2%). Further, RNAseq data (Fig. 3c) showed low levels of *CTNNA1* expression in the

PDOs derived from CSF DTCs in the two cases with focal *CTNNA1* deletions (KCL622 and KCL625).

8: The authors say "CTNNA1 and CDH1 aberrations displayed mutually exclusivity in BCLM samples". Is this statistically significant?

Response: Using the data below, a Fisher's exact test was performed showing a statistically significant absence of association (Odds Ratio 0.00, $p = 0.0351$, two-sided test). This is now stated in the Results section of the revised manuscript.

		CDH1	
		mut/del	wt
CTNNA1	mut/del	0	4
	wt	11	6

9: 8p loss is fairly common in breast cancer? From what is presented, it is difficult to see that *ARHGEF10* or *DMTN* are the genes of interest in 8p? How focal are these events?

Response: We acknowledge that 8p loss is a common event in breast cancers (as indicated on p.5, final paragraph) in addition to other arm-level copy number losses as noted by reviewers (e.g. 16q). For this reason, we restricted the definition of genetic 'alterations' for biological pathway analysis, to include only high-level copy number events i.e. deep deletions or amplifications, rather than losses and gains.

As a consequence, although copy number loss of *ARHGEF10* was common (discovered in 13 CSF samples), only 4 of these were deep deletion events. Deep deletion events comprised one focal event (1.9 Mb) and 3 broader, but not arm-level, events (6.7 – 6.9 Mb). The breadth of copy number alterations can be viewed at a sample level in the **new Supplementary Table 6**. *ARHGEF10* alteration rate (one mutation plus four deep deletions) in BCLM cohort was significantly higher (5/21) than the MBC cohort (7/216) (p value 0.000339; as shown in Supplementary Table 8), therefore we have flagged this as a gene of interest. The text in result section has been updated to describe these findings in better detail highlighting the rationale for consider *ARHGEF10* alterations as not merely passenger events. Page 7, final paragraph.

Interestingly, a member of the RhoGEF family, *ARHGEF10* **was frequently** aberrated in BCLM CSF cfDNA (23% of cases), comprising four deep deletions **(one focal event ~1.9 Mb and 3 broader, but not arm-level events between 6.7 – 6.9 Mb) (Supplementary Table 6)** and a BCLM-unique missense mutation (p.A1100P) **with predicted pathogenicity by SIFT, Polyphen2 and CADD (Supplementary Table 3).**

We identified *DMTN* as a gene of interest, since three deep deletions and three mutations were discovered in CSF cfDNA. The deep deletions events were broad rather than focal, however combined with the finding of deleterious mutations in three cases, points to the possibility that loss or defective *DMTN* is a potential driver aberration in BCLM. As described in the next response, we have updated the text to explain the rationale and caveats to this finding.

10: A lot of missense mutations are reported in the section titled "Biological pathway analysis shows enrichment for cytoskeletal aberrations in BCLM". For example, for *DMTN*, *SPTA1*. How do we know if these mutations are deleterious or of any functional relevance?

Response: New Supplementary Table 3 lists all somatic variants discovered by WES, and we have annotated each variant with mutation effect prediction scores for nonsynonymous variants: SIFT, PolyPhen2 HVAR, and CADD phred score.

Three mutations with predicted pathogenic effect were discovered in *DMTN*. Two of the mutation events were frameshift indels (p.P94Hfs*137 and p.P95Tfs*38) with possibility of leading to a truncated protein. The *DMTN* nonsynonymous variant (p.R268C) scored highly on the three mutation effect predictors (Polyphen2_HVAR_score = 0.988, SIFT score = 0.007, CADD phred score = 35) indicating this amino acid change as likely deleterious. This gene was not profiled in the MBC cohort; however it has a low alteration rate (4%) in breast cancers profiled for TCGA (cBioportal). Further, when looking at matched tissues sequencing (as shown in Supplementary Fig. 5) *DMTN* deep deletion and mutations can be seen to mostly occur in CSF samples rather than the matched tissues. Therefore *DMTN* mutation/deletion appears to be a BCLM associated alteration rather than a general breast cancer variant.

Spectrin-alpha 1 (*SPTA1*), a scaffold protein which links the plasma membrane to the actin cytoskeleton, was commonly aberrated by missense variants (4/21). Combined Annotation-Dependent Depletion (CADD) phred scores were greater than 10 for three of the four missense *SPTA1* mutations (p.R739H, p.K2352T and p.D1421H). Germline missense variants in *SPTA1* lead to hereditary spherocytosis, a condition leading to abnormal rounding of red blood cells due to disturbance in association between the cytoskeleton and the overlying lipid bilayer. Pathogenic variants leading to hereditary spherocytosis can occur at multiple positions in the *SPTA1* gene (van Vuren et al., 2019) Although an erythroid cell protein, inspection on The Human Protein Atlas shows that *SPTA1* is expressed in numerous solid tumours including breast cancer. Therefore there is a potential that *SPTA1* mutations could have relevance in cancer biology, as suggested in colorectal cancer (Palaniappan et al., 2016) and small cell lung cancer (Iwakawa et al., 2015) studies. The relevant section of the results now states

Myosin 15A (*MYO15A*), encoding an actin-based motor molecule, was aberrated in **6/21 CSF cfDNA samples (one truncating mutation and four missense mutations with predicted pathogenicity), which were often accompanied by loss-of-heterozygosity (LOH) (Fig. 3b; Supplementary Table 3)**. Dematin actin binding protein (*DMTN*), a regulator of cytoskeleton remodelling (Lutchman et al., 2002; Mohseni and Chishti, 2008) **was frequently aberrated in CSF cfDNA (Fig. *DMTN* downregulation has been reported to promote colorectal cancer metastasis through activation of Rac1, a key cytoskeletal regulator (Lutchman et al., 2002; Mohseni and Chishti, 2008). Although the *DMTN* copy number events were part of broader 8p deletion events (18 - 29 Mb), the finding of deleterious frameshift and missense mutations in a further 3 three CSF cfDNA, which were more often BCLM-private mutations, raises the possibility that loss or defective *DMTN* could be a potential driver aberration in BCLM.** Spectrin-alpha 1 (*SPTA1*), a scaffold protein linking the plasma membrane to the actin cytoskeleton **and predominant cause of hereditary**

spherocytosis due to dysregulated erythroid cell shape when mutated in germline DNA (van Vuren et al., 2019), was aberrated in 4/21 BCLM CSF cfDNA samples by 3b). somatic missense mutations, of which three had high pathogenicity scores using CADD predictions (Fig. 3b; Supplementary Table 3).

11: In section "BCLM subclones seed early during primary tumour evolution", the authors say their analyses revealed "(a) rather than a late-metastatic seeding event, BCLM disseminates early from the primary tumour, (b) CSF and plasma cfDNA display divergent evolution, often sharing branches with other metastatic sites but possessing distinct mutational repertoires, ". How many cases of 21 patients fall into these? These should clearly be indicated in text and in **Figure 3c and S. Figure 4**. The way they are the figures are very difficult to follow.

Response: We thank the reviewer for highlighting this point. In the revised manuscript we have supplied an additional table (**new Supplementary Table 9**) describing the evolution patterns for 20 cases where phylogenetics was carried out. In addition, we have added the following detail in the relevant text on p.9 final paragraph.

To investigate the **clonal** evolution of BCLM, **variant allele frequencies and allele-specific copy number states were used to infer mutational subclones and compose phylogenetic trees (Fig. 3c; Supplementary Fig. 6). Within the limitation of WES, which may underestimate subclonal composition compared to whole genome sequencing, these analyses revealed (a) BCLM dissemination occurs early from the primary tumour rather than as a late-metastatic seeding event in 16 BCLM cases as described in Supplementary Table 9, (b) CSF and plasma cfDNA display divergent evolution, often sharing branches with other metastatic sites but possessing distinct mutational repertoires; this occurred in 7/11 cases with matched CSF and plasma samples, implying that there was ongoing evolution at the BCLM and extracranial sites, whereas a linear evolution between CSF and plasma occurred in 4/11 cases, implying BCLM and extracranial metastases may have derived from the same metastatic seeding event,**

12: The authors say "Furthermore, two CSF samples, KCL566 and KCL148, exhibited high TMB (>10 mutations/Mb) (Fig. 2) predictive of response to immune checkpoint inhibition. It was noted that both these TMB-high cases had truncal MLH1 aberration however substantial elevation of TMB was seen in CSF cfDNA compared to matched tissues indicating further accumulation of mutational events in the leptomeninges". Could the increase in TMB in the CSF be because multiple distinct clones with a high TMB are found and detected in the CSF?

Response: The phylogenetic trees in Fig. 3c and Supplementary Figure 6 show the subclone evolution in these two cases, and the mutation counts of these subclones. Both KCL566 and KCL148 CSF cfDNA were composed of only two additional subclones, therefore we did not conclude that a high TMB was associated with multiple distinct subclones. A limitation of our phylogenetic data, however, is that subclone analysis was performed using WES rather than WGS data, therefore may underestimate the subclone composition of the samples. We have inserted a statement to acknowledge this caveat (please see above in Response 11).

13: In section "Development and characterization of BCLM PDOs" the authors say "The high concordance between CSF cfDNA and CSF derived tumour cell DNA, confirms that CSF cfDNA represents a valid surrogate for the rarely available BCLM genetic and biological

material.". Compared to what? What is the overlap with other samples available for these 5 patients.

Response: In response to this helpful suggestion, we have compared the CNA and mutational concordance between BCLM PDOs and matched primary tumour and for all 5 models. We have displayed these in a new figure (**new Supplementary Fig. 7**), along with comparisons between BCLM PDOs and matched plasma (n = 2) and metastatic sites (n = 1). We have used the concordance values between PDOs and CSF or primary tumour, to determine that PDOs more closely match CSF cfDNA than primary tumour (shown with significance values, in a new figure panel (**new Fig. 4c**).

The text has been appropriately update to reflect these findings (page 11, paragraph 2).

Following WES, a comparison of CNAs between the BCLM PDOs and their matched CSF cfDNA samples revealed a high level of concordance (Fig. 4b upper panel) while comparison of the mutational landscape revealed a median of 81.5% of CSF cfDNA detected variants were present in their matched BCLM PDOs (Fig. 4b lower panel). Comparison between BCLM PDOs and primary tumours revealed a lower concordance in CNAs/mutations (Fig. 4c; Supplementary Fig. 7) indicating that CSF cfDNA represents a valid surrogate for the rarely available BCLM genetic and biological material.

14: The authors note lower expression of CDH1 and CTNNA1 in PDOs. How were these genes identified? Is this significant when differential expressed genes are identified in an unbiased manner?

Response: Unfortunately, we generated the primary and PDOs RNA datasets on different sequencing technologies in two different batches. Given RNA is highly sensitive to batch/sequencing effects, it is challenging to create a list of differentially expressed genes using formal statistics e.g. adjusting the model for batch, which in this case is also the factor/variable of interest (specimen: primary vs PDO). While we could fit such a model (or explore batch effect correction) and perform differential gene expression analysis, it would be hard to distinguish between the true biological difference due to the specimen (primary, PDO) and technical batch. We have therefore taken a careful and exploratory approach where we have limited the comparison to key adherens junction genes only without formal statistics (Fig. 4d).

The RNAseq raw data have been deposited in SRA (SUB13503867) and are publicly available.

Reviewer #2 (Remarks to the Author): expertise in bioinformatics analysis of genomic evolution

The manuscript by Fitzpatrick et al, describes a rare event in breast cancer, i.e. breast cancer leptomeningeal metastases (BCLM). The authors study the inter-tumoral heterogeneity between BCLM (CSF cfDNA), extracranial metastases (plasma cfDNA or metastatic tissue) and the primary tumour, putting together a unique compendium of data from these sites. Furthermore, the authors manage to culture patient derived organoids (PDOs) from disseminated tumor cells in the CSF, which they could show have similar mutational spectra as those found in the cfDNA, which is remarkable. Following WES the authors compare the copy number profiles between the PDOs and their matched CSF cfDNA samples and saw high level concordance and around 81.5% of the detected variants were present in both the cfDNA and the organoids. This is a spectacular result as it shows that CSF cfDNA represents a valid surrogate for the rarely available BCLM genetic and biological material. I am missing here a little comparison of the mutational spectre in the PDOs to that of the primary tumor.

Response: We thank the reviewer for these comments. With regards the comparison of the PDOs to the primary tumours, in the revised manuscript these comparisons are provided in new Supplementary Figure 7 and summarised in new Fig 4c.

Comments

1: Loss of E-cadherin staining and reduced CDH1 expression was seen in one PDO by Immunohistochemical staining and transcriptional profiling of the PDOs and matched primary tumours showed specific differences as loss of membrane associated α -catenin staining and reduced CTNNA1 expression in other PDOs. However, it is known that PDOs differ substantially from primary tumors and much of these differences are due to culturing conditions. In this paper, the authors interpret the observed in the PDO qualities as directly characteristics of the DTCs, which is extrapolation.

Response: We agree that difference in expression and protein levels can occur as a results of culture conditions (see below regarding our response to the question of ER expression). However, for the case of E-cadherin and CTNNA1 these statements are supported by the WES data from PDOs where we show (a) for KCL4566 a *CDH1* truncating mutation on Exon 4 (p.L139X) with a variant allele frequency of 98%. In keeping with the high mutant VAF, copy number data showed a loss of heterozygosity (\log_2 ratio of -0.801). (b) both PDOs KCL622 and KCL625 display copy number loss of *CTNNA1*. Therefore the detected genomic changes in *CDH1* and *CTNNA1* occurring in the BCLM cells (cultured as PDOs), are translated to the transcriptional and translational level.

We have update the relevant section of Results as follows:

Consistent with genomic analysis of CSF cfDNA (Fig. 1-3), immunohistochemical staining (Fig. 4a) of PDOs and matched primary tumours (Table 1) revealed a reduction of E-cadherin staining in PDO KCL566 (IC-NST with acquired *CDH1* truncating mutation in CSF cfDNA and BCLM PDO detected by WES) commensurate with the reduced *CDH1* expression compared to its primary tumour (RNAseq; Fig. 4d), the lack of *CDH1* expression detected by RTqPCR (Fig. 4e) and the lack of E-cadherin staining when grown as patient-derived xenografts (PDXs; see later).

2: Also notably the loss of ER expression has been observed multiple times when culturing primary tumors, and experimental conditions need to be carefully controlled. Can the authors show that when they culture ER positive tumors with these conditions, the ER status is preserved?

Response: As we state in our manuscript loss of ER expression is a common problem when establishing PDOs from ER+ tumour samples (Li et al., 2022). Using the Ocello primary organoid medium supplemented with oestradiol we have established an ER+ PDO from a human breast cancer that had disseminated to the peritoneum (ascites). The original patient sample was inoculated intraperitoneally into mice with oestradiol supplementation. The ascitic cell population that developed was then established as a PDO culture and retained ER expression. However, this patient did not develop BCLM and therefore was not part of this study.

Minor comments:

3: I read easily the figures and the legends to figures, which I find clear and well illustrating the experimental flow. The text of the paper is however full of details that obscure the main message, and the comparisons made.

Response: We apologise. In the revised manuscript we have endeavoured to describe more clearly our findings throughout this manuscript and to remove unnecessary detail.

4: Some wording:

In "Consequently although 21/21 of CSF cfDNA samples had sufficient tumour content only 11/21 paired plasma cfDNA samples were suitable for WES.»

Replace «sufficient» with «detectable», not clear sufficient for what

Response: This statement has been rewritten in the revised manuscript to clarify that 'sufficient' referred to $\geq 10\%$ tumour content.

Reviewer #3 (Remarks to the Author): expertise in breast cancer brain metastasis modelling

Fitzpatrick et al. describes genomic and pre-clinical modeling of breast cancer leptomeningeal metastases (BCLM), revealing that BCLM acquires a “lobular-like” phenotype. The authors have assembled a cohort of 21 patients with BCLM, from which cell free DNA (cfDNA) from CSF and plasma (subset of 11 patients) were available. Tissue from primary breast tumors and extracranial metastases (when available) was also analyzed. Whole exome sequencing (WES) was performed on the cfDNA samples (CSF and plasma), primary tumors and extracranial metastases. Importantly, the authors have established patient-derived organoids (PDOs) from cells purified from CSF and characterized these models for their ability to form leptomeningeal metastases in mice. The authors conclude that BCLMs exhibit an enrichment of specific mutations (e.g. CDH1, ARHGEF10, CTNNA1) relative to primary breast tumors. They also highlight the observations that many mutations observed in cfDNA from CSF implicate genes that encode constituents of cellular junctions, regulators of cytoskeletal architecture and modulators of cell migration/invasion.

The study of leptomeningeal metastases is an important area of research that requires concerted efforts to develop models to better understand the molecular underpinnings of these lesions. The authors should be congratulated for this informative and well written manuscript that advances our understanding of leptomeningeal metastases. However, there are several points that the authors can address to further strengthen the manuscript.

Main Points:

1: The authors use **strong language throughout** the manuscript such as, but not limited to “We found that BCLM samples, whether derived from ILC or non-lobular primary tumours, were enriched for alterations in CDH1”. When examining the data presented in **Figure 2 and Supplemental Figure 2**, of the N=7 primary breast tumors included in the analyses that did not have CDH1 alterations at baseline, only 2 gained such an alteration in the specimen collected from CSF (1 loss, 1 stop-gain mutation). Furthermore, when looking at CTNNA1, of N=13 patients who had no CTNNA1 alterations in the primary tumor, CTNNA1 deletion was identified in 2 matched CSF specimens. In one primary tumor where CTNNA1 was lost, it was no longer lost in the CSF specimen. Ideally, a higher number of specimens should be collected to develop a statistically-based argument to substantiate these claims. If this is not feasible, the authors should qualify the statements made throughout the manuscript.

Response: Throughout the revised manuscript we have 'toned' down our language, stated caveats to conclusions drawn where relevant and endeavoured to present the data more clearly and succinctly. We agree that a relatively small number of BCLM cases were collected and analysed in this study. That said, the most notable feature of these samples was (a) within the cohort of patients within the study there was a notable enrichment for patients who had a primary diagnosis of lobular breast cancer (provide numbers), and (b) that with the patient who had a primary diagnosis of IC-NST (n=xxx) that 55% had acquired mutations in CDH1 or CTNNA1 in the leptomeningeal sites - hence our conclusion that having or acquiring a lobular or lobular-like phenotype may predispose breast cancer cells from homing to, or flourishing within, the leptomeningeal space.

2: The OncoKB analysis where the authors claim that 57% of patients acquired therapeutic Level 1 alterations in the CSF specimens appears to be overstated. For example, while

PIK3CA E542K and H1047R mutations are indeed described as **Level 1 alterations**, other PIK3CA mutations such as N1044K or E39K mutations are not listed in OncoKB's database. Other examples include ATM, for which mutations only have Level 1 evidence in prostate cancer, or BRCA2 mutations which are listed as Level 3A mutations in breast cancer. These analyses should be revised to accurately reflect the true actionability of these newly identified mutations.

Response: In the revised manuscript we provide a revised version on the OncoKB table (**revised Supplementary Table 10**) where we display the CSF cfDNA variants with OncoKB 'actionability' as described with a Therapeutic Level between 1 – 4. These therapeutic levels were obtained from the OncoKB database on 29th April 2022, and we noted the highest therapeutic level for the specific alteration in any cancer type. We chose not to limit this to breast cancer given that large scale mutation-informed interventional studies such as NCI-MATCH (Flaherty et al., 2020) allocate treatments across multiple solid tumours, using therapies that have shown clear evidence of clinical benefit or at least promising preliminary efficacy in any tumour histology for the identified aberration. However, we acknowledge that the level of actionability in a breast cancer setting can be lower and the treatment of aberrations is always context dependent. Therefore, a **new column** has been added to Supplementary Table 10 listing the therapeutic level for 'any cancer' and the therapeutic level in 'breast cancer' alongside the column for 'any cancer'.

In relation to specific alterations raised by the reviewer, the non-hotspot PIK3CA mutations N1044K (PI3/4-kinase catalytic domain) or E39K (PI3K p85 regulatory subunit binding domain) have been annotated in the OncoKB curated list 'PIK3CA Oncogenic Mutations' as Therapeutic Level 2. In our original manuscript we have updated Supplementary Table 10 for these variants to reflect this. Other non-hotspot PIK3CA mutations were omitted from Supplementary Table 10 since they are not described as 'actionable' by OncoKB.

In addition to revising Supplementary Table 10 as described, we have updated the text with the following clarifications (p.10, final paragraph)

Including actionable variants found only in CSF cfDNA samples, and shared with other tissues, alterations with potential therapeutic actionability (OncoKB therapeutic alterations level 1 - 4) were discovered in 17/21 CSF cfDNA samples, and were private to CSF in 4/21 (potentially actionable variants highlighted in green in Fig. 3c, and Supplementary Fig. 6, and listed with OncoKB therapeutic levels in Supplementary Table 10). Those occurring in multiple cases being *ARID1A* deleterious mutations in 3 cases, *PIK3CA* activating mutations in 5 cases, *PTEN* deleterious mutations in 2 cases and *MDM2* amplifications in 3 cases. Although this reveals the potential for improved targeting of BCLM using therapy tailored to the identified genomic aberrations, **important caveats to interpreting the potential actionability of identified variants are (a) the uncertainty of drug penetration through the blood-brain and blood-CSF barriers to reach therapeutic concentrations within the CSF, and (b) limited licensed indications for agents that target the biology associated with these variants in a breast cancer setting.**

3: In **Figure 4A**, the authors claim that E-cadherin expression is lost by **IHC (Fig. 4a)** and RNAseq (**Fig. 4c**) analysis when comparing between the primary mammary tumor (positive) and the KCL566 PDO (negative). The RNAseq data does suggest a significant downregulation

of CDH1 levels; however, the IHC image presented in the Figure appears to show some membranous E-cadherin staining. When injected into the mammary fat pad, the KCL566 PDO gives rise to E-cadherin negative mammary tumors (**Fig. 5a**). Can the authors comment on this apparent variability?

Response: We agree with the reviewer that the IHC staining of the KCL566 organoid shows some membranous staining. We have investigated this in our laboratory, and we believe that this reflects some cross reactivity of the antibody with P-cadherin. Whatever the reason, we are confident that the KCL566 PDO is E-cadherin negative based on (a) whole exome sequencing which shows a *CDH1* truncating mutation on Exon 4 (p.L139X) with a variant allele frequency of 98%. In keeping with the high mutant VAF, copy number data showed a loss of heterozygosity (\log_2 ratio of -0.801). The same truncating mutation and LOH was present in CSF cfDNA, but no other samples from this patient. (b) we have performed RTqPCR analysis of the 5 PDOs, plus two E-cadherin +ve cell lines (MCF7 and T47D) and two E-cadherin -ve cell lines (MDA-MB-134 and SUM44PE) as controls. These new data (**new Fig. 4d**) show that PDO KCL622 expresses *CDH1* at levels equivalent to the positive cell line controls while the other 4 PDOs are *CDH1*-low (KCL450, KCL625) or negative (KCL320, KCL566).

4: In the analyses presented in **Figures 5 and 6**, the authors describe differences between dural, pituitary, brain and leptomeningeal metastases. The authors are encouraged to include **representative images** demonstrating these various metastatic lesions from the 5 PDO models in the revised manuscript (from the cardiac-injected cohorts – Fig. 6A). Additional text providing more detailed methodology, particularly for the identification and determination of pituitary metastases, would be helpful. Are the pituitary metastases that develop following intra-cardiac injection of the PDO models similar to the skull-based lesions described from leptomeningeal metastasis models of medulloblastoma (Garzia et al. Cell, 2018)?

Response: As requested we have now included example images of the dural and pituitary metastases to go alongside the images of brain and leptomeningeal metastases (**revised Fig. 6a** (lower panel)). A consultant histopathologist (I.R.) provided independent review of these images and as now stated in the Methods, confirmed the identification of pituitary and dural metastases.

The reviewer asks whether these pituitary metastases are similar to the skull-based lesions described by (Garzia et al., 2018). We have read the Garzia manuscript carefully, and other manuscripts from the lead authors, and can no reference to skull-based medulloblastoma lesions in their mouse models. Consequently, we cannot answer this question.

5: The authors provide a careful characterization the metastases that are formed following mammary fatpad and intra-cardiac injections for all 5 PDO models. However, only the KCL320 model was injected intracerebroventricularly. Do all 5 PDOs form only leptomeningeal metastases following this route of injection? For example, is there any difference in the ability to form LM using models that have no (KCL320, KCL450), low (KCL566) or high (KCL622 and KCL625) E-Cadherin levels? For the KCL320 model, what is the penetrance it of mice that develop LM when the cells are injected intracerebroventricularly (appears to be 50%: **Fig. 6b**).

Response: In this study only KCL320 was inoculated intracerebroventricularly. The reason for this was that KCL320 did not grow when injected orthotopically, intraductally or intraperitoneally (Fig. 5) but via the intracardiac route 100% of the mice had tumours cells in the brain parenchyma and/or leptomeninges - suggesting that this model might have a strong predilection for the CNS (Fig. 6a). Consequently, we injected KCL320 ICV into 4 mice (Fig.

6b). Of the 4 mice, two developed leptomeningeal disease, one where the disease was restricted to the cranial meninges and one where disease was seen in both the cranial and spinal meninges. Apologies, for not including the take rate in these mice in the original manuscript, this is now stated in the Results section and figure legend.

6: An open question in the field of LMD is whether cells floating in the CSF are true proxies for the disease, or whether these cells are simply the ones not able to survive and proliferate in contact with the arachnoid/pia mater. **Remsik et al. 2022 (Cancer Rep.)** has published data beginning to solve this question, associating floating cells with aggressivity in the context of LMD. In relation to comment 5 above – do the 5 PDO models, when injected intracerebroventricularly, give rise to LM that shed cells into the CSF to varying degrees? The morphologies of the organoids are quite different, with some forming compact organoids (KCL320, KCL566, KCL625), slightly diffuse (KCL450) and very diffuse (KCL622) organoids. Would this relate to the extent of LMD in the various PDOs? It seems that these models would be very useful in this regard. It is interesting that the KCL622 model seems to generate the greatest degree of metastases following cardiac injection; whereas the KCL566 model produces more spontaneous metastases.

Response: As we only injected KCL320 via the ICV route we cannot answer this question directly. However, the referee raises a very interesting point. We have now referenced the (Remsik et al., 2022) publication in support of the viability of floating cells in the CSF. Of note - the difficulty in growing cells from the CSF may reflect the high incidence of lobular and lobular-like breast cancers colonising this space as it is known that lobular breast cancers, particularly those with classical ILC features, are problematic to establish in *in vitro* culture. As detailed in our response to Reviewer 4 below, we have expanded the methods section to provide more information on the PDO culture conditions, including the use of meningeal cell conditioned medium when establishing the PDO cultures.

7: The authors provide intriguing data suggesting that ROS1 inhibition may be a useful therapeutic strategy for LMD with this lobular-like phenotype. This notion has important translational implications and would benefit by being further elucidated. Have the authors assessed expression of ROS1 in their PDO models? This would be interesting data to show and subsequently correlate sensitivity of the PDO models to ROS levels. The *in vitro* sensitivity of the PDO models is interesting; however, it would be important to assess responsiveness to ROS1 inhibition *in vivo* with several of these models. If possible, a model of LMD that has high and low/no expression of ROS1, but still derives from a lobular/CDH1-mutated breast cancer, would be worthwhile studying.

Response: We thank the referee for this question. In their original publication, Bajrami and colleagues reported a sensitivity of E-cadherin-defective breast cancer models to ROS inhibitors (Bajrami et al., 2018). In that study, the authors used both crizotinib and foretinib. In our manuscript we used crizotinib and taletrectinib as the latter is blood-brain-barrier penetrant. All three inhibitors are not specific for ROS1. The best studied, crizotinib, has been described as a multikinase inhibitor, binding to 13 out of 178 kinases tested (Vasta et al., 2018) and classed as a potent inhibitor of ROS1, MET and ALK. Taletrectinib, is listed as a potent inhibitor of ROS1 and all three members of the NTRK family.

To address the reviewer's question, we examined the PDO RNAseq data for expression of ROS1, MET, ALK and NTRK family members (see below). Interestingly, the PDO with least response to the two inhibitors, PDO KCL450, showed negligible expression of ALK and

NTRK1. However, given that all 5 PDOs presented in this manuscript have a lobular or lobular-like features, a larger panel of PDOs - including non-lobular PDOs - would be required to draw any firm conclusions. That said, we should have been more accurate in our description of these inhibitors and this section of the manuscript has been revised.

Reviewer #4 (Remarks to the Author): expertise in development of PDO models

In the manuscript presented for review, “Genomic profiling and pre-clinical modelling of breast cancer leptomeningeal metastasis (BCLM) reveals acquisition of a lobular-like phenotype,” Fitzpatrick and colleagues report the whole exome sequencing of breast tumors in patients with leptomeningeal metastases, along with sequencing of matched cell-free DNA from cerebrospinal fluid (CSF circulating tumor DNA) and serum (circulating tumor DNA), and DNA isolated from other metastatic lesions. The researchers compare the mutational profiles from each specimen type (e.g., tumor DNA from CSF vs. plasma) to explore the mutational spectrum and different evolutionary pressures, such as from chemotherapeutic treatment. This is a well-controlled study with added power from the analysis of the various tumor sites (primary, metastatic, and circulating DNA). Furthermore, the authors report development of primary ex vivo cell models (3D matrigel embedded) established from circulating CSF tumor cells that closely mimicked the phenotype of matched primary tumors. Furthermore, xenografts were successfully generated from the 3D organoids (injected into NSG mouse mammary fat pads), or dissociated cells into cardiac ventricles or cerebral ventricles that permitted analysis of growth, metastatic spread, and testing of chemotherapeutic agents. The authors correctly describe the grim prognosis of patients with leptomeningeal metastasis, and the findings and models they describe here are indeed an important resource for studying the biology of these rare tumors. The results contribute important information to the paucity of that surrounding this rare but highly lethal condition. This reviewer supports the publishing of this manuscript with only very minor revisions.

Suggestions:

1- On first read, I found the abstract a bit strong on lingo and it was a bit confusing to understand what the researchers had done and what this article contained. It needs minor refinement.

Response: Apologies - we have revised the abstract and endeavoured to describe more clearly our findings throughout this manuscript

2- In the introduction, when describing lobular breast cancer, it would be helpful if the authors mentioned that the loss of E-cadherin is a general distinguishing characteristic of lobular breast cancers, as this is important for understanding the mutational profile of CDH1 (Fig. 1e). The authors discuss this later, but it would be helpful here.

Response: This information is now included in the Introduction

3- Authors need to define PDOs and give critical details of their generation in the narrative, as these can be generated in a variety of different ways.

Response: PDO is now defined. The methodology for PDO generation is fully described in the Methods section (including details for the Ocello medium - see below). In the results section we have provided the essential details of the methodology.

4- What is Ocello primary cell medium? Is it proprietary? What are the key components that distinguish this medium?

Response: At the time of performing this study, the Ocello primary organoid medium was proprietary. However, Ocello is now part of Crown Bioscience they have provided us with a full list of components.

The medium provided by Ocello has the following composition:

Advanced DMEM/F12

10 mM HEPES

1× Glutamax

1× B-27 without retinoic acid

1× N-2

20 ng/ml FGF2 (fibroblast growth factor 2)

50 ng/ml EGF (epidermal growth factor)

10 μM Y-27632

5 μM A83-01

1.25 mM N-acetylcysteine

1x penicillin and streptomycin

As described in the Methods, in our study the Ocello media was additionally supplemented with

10 ng/mL neuregulin-1

10 ng/mL EGF

17β-oestradiol (10^{-11} M) - for ER+ breast cancers.

During the initial establishment of the organoids (prior to their first passage) we mixed the Ocello supplemented medium' 1:1 with meningeal cell conditioned medium (referred to as PDO culture medium). The full composition of the media and the method for generating meningeal cell conditioned medium is now included in the Methods section

In summary, in terms of what components distinguish this medium

- (a) The Ocello medium was based on an organoid culture medium described by (Calon et al., 2015) with the following changes: addition of 1.25 mM N-acetylcysteine, addition of 1x penicillin and streptomycin, removal of 1 μM LY2157299 and replacement with an alternative ALK inhibitor, 5 μM A83-0.
- (b) we supplemented the Ocello medium with 10 ng/mL neuregulin-1, 10 ng/mL EGF with/without 10^{-11} M 17β-oestradiol.
- (c) we included meningeal cell conditioned medium during establishment of the PDOs

References

- Alexandrov, L.B., J. Kim, N.J. Haradhvala, M.N. Huang, A.W. Tian Ng, Y. Wu, A. Boot, K.R. Covington, D.A. Gordenin, E.N. Bergstrom, S.M.A. Islam, N. Lopez-Bigas, L.J. Klimczak, J.R. McPherson, S. Morganella, R. Sabarinathan, D.A. Wheeler, V. Mustonen, P.M.S.W. Group, G. Getz, S.G. Rozen, M.R. Stratton, and P. Consortium. 2020. The repertoire of mutational signatures in human cancer. *Nature*. 578:94-101.
- Bajrami, I., R. Marlow, M. van de Ven, R. Brough, H.N. Pemberton, J. Frankum, F. Song, R. Rafiq, A. Konde, D.B. Krastev, M. Menon, J. Campbell, A. Gulati, R. Kumar, S.J. Pettitt, M.D. Gurden, M.L. Cardenosa, I. Chong, P. Gazinska, F. Wallberg, E.J. Sawyer, L.A. Martin, M. Dowsett, S. Linardopoulos, R. Natrajan, C.J. Ryan, P.W.B. Derksen, J.

- Jonkers, A.N.J. Tutt, A. Ashworth, and C.J. Lord. 2018. E-Cadherin/ROS1 Inhibitor Synthetic Lethality in Breast Cancer. *Cancer Discov.* 8:498-515.
- Brastianos, P.K., S.L. Carter, S. Santagata, D.P. Cahill, A. Taylor-Weiner, R.T. Jones, E.M. Van Allen, M.S. Lawrence, P.M. Horowitz, K. Cibulskis, K.L. Ligon, J. Tabernero, J. Seoane, E. Martinez-Saez, W.T. Curry, I.F. Dunn, S.H. Paek, S.H. Park, A. McKenna, A. Chevalier, M. Rosenberg, F.G. Barker, 2nd, C.M. Gill, P. Van Hummelen, A.R. Thorner, B.E. Johnson, M.P. Hoang, T.K. Choueiri, S. Signoretti, C. Sougnez, M.S. Rabin, N.U. Lin, E.P. Winer, A. Stemmer-Rachamimov, M. Meyerson, L. Garraway, S. Gabriel, E.S. Lander, R. Beroukhim, T.T. Batchelor, J. Baselga, D.N. Louis, G. Getz, and W.C. Hahn. 2015. Genomic Characterization of Brain Metastases Reveals Branched Evolution and Potential Therapeutic Targets. *Cancer Discov.* 5:1164-1177.
- Calon, A., E. Lonardo, A. Berenguer-Llargo, E. Espinet, X. Hernando-Mombona, M. Iglesias, M. Sevillano, S. Palomo-Ponce, D.V. Tauriello, D. Byrom, C. Cortina, C. Morral, C. Barcelo, S. Tosi, A. Riera, C.S. Attolini, D. Rossell, E. Sancho, and E. Batlle. 2015. Stromal gene expression defines poor-prognosis subtypes in colorectal cancer. *Nat Genet.* 47:320-329.
- Ciriello, G., M.L. Gatz, A.H. Beck, M.D. Wilkerson, S.K. Rhie, A. Pastore, H. Zhang, M. McLellan, C. Yau, C. Kandoth, R. Bowlby, H. Shen, S. Hayat, R. Fieldhouse, S.C. Lester, G.M. Tse, R.E. Factor, L.C. Collins, K.H. Allison, Y.Y. Chen, K. Jensen, N.B. Johnson, S. Oesterreich, G.B. Mills, A.D. Cherniack, G. Robertson, C. Benz, C. Sander, P.W. Laird, K.A. Hoadley, T.A. King, T.R. Network, and C.M. Perou. 2015. Comprehensive Molecular Portraits of Invasive Lobular Breast Cancer. *Cell.* 163:506-519.
- Curtis, C., S.P. Shah, S.F. Chin, G. Turashvili, O.M. Rueda, M.J. Dunning, D. Speed, A.G. Lynch, S. Samarajiwa, Y. Yuan, S. Graf, G. Ha, G. Haffari, A. Bashashati, R. Russell, S. McKinney, M. Group, A. Langerod, A. Green, E. Provenzano, G. Wishart, S. Pinder, P. Watson, F. Markowitz, L. Murphy, I. Ellis, A. Purushotham, A.L. Borresen-Dale, J.D. Brenton, S. Tavare, C. Caldas, and S. Aparicio. 2012. The genomic and transcriptomic architecture of 2,000 breast tumours reveals novel subgroups. *Nature.* 486:346-352.
- Desmedt, C., G. Zoppoli, G. Gudem, G. Pruneri, D. Larsimont, M. Fornili, D. Fumagalli, D. Brown, F. Rothe, D. Vincent, N. Kheddoumi, G. Rouas, S. Majjaj, S. Brohee, P. Van Loo, P. Maisonneuve, R. Salgado, T. Van Brussel, D. Lambrechts, R. Bose, O. Metzger, C. Galant, F. Bertucci, M. Piccart-Gebhart, G. Viale, E. Biganzoli, P.J. Campbell, and C. Sotiriou. 2016. Genomic Characterization of Primary Invasive Lobular Breast Cancer. *J Clin Oncol.* 34:1872-1881.
- Flaherty, K.T., R.J. Gray, A.P. Chen, S. Li, L.M. McShane, D. Patton, S.R. Hamilton, P.M. Williams, A.J. Iafrate, J. Sklar, E.P. Mitchell, L.N. Harris, N. Takebe, D.J. Sims, B. Coffey, T. Fu, M. Routbort, J.A. Zwiebel, L.V. Rubinstein, R.F. Little, C.L. Arteaga, R. Comis, J.S. Abrams, P.J. O'Dwyer, B.A. Conley, and N.-M. team. 2020. Molecular Landscape and Actionable Alterations in a Genomically Guided Cancer Clinical Trial: National Cancer Institute Molecular Analysis for Therapy Choice (NCI-MATCH). *J Clin Oncol.* 38:3883-3894.
- Garzia, L., N. Kijima, A.S. Morrissy, P. De Antonellis, A. Guerreiro-Stucklin, B.L. Holgado, X. Wu, X. Wang, M. Parsons, K. Zayne, A. Manno, C. Kuzan-Fischer, C. Nor, L.K. Donovan, J. Liu, L. Qin, A. Garancher, K.W. Liu, S. Mansouri, B. Luu, Y.Y. Thompson, V. Ramaswamy, J. Peacock, H. Farooq, P. Skowron, D.J.H. Shih, A. Li, S. Ensan, C.S. Robbins, M. Cybulsky, S. Mitra, Y. Ma, R. Moore, A. Mungall, Y.J. Cho, W.A. Weiss,

- J.A. Chan, C.E. Hawkins, M. Massimino, N. Jabado, M. Zapotocky, D. Sumerauer, E. Bouffet, P. Dirks, U. Tabori, P.H.B. Sorensen, P.K. Brastianos, K. Aldape, S.J.M. Jones, M.A. Marra, J.R. Woodgett, R.J. Wechsler-Reya, D.W. Fults, and M.D. Taylor. 2018. A Hematogenous Route for Medulloblastoma Leptomeningeal Metastases. *Cell*. 173:1549.
- Iwakawa, R., T. Kohno, Y. Totoki, T. Shibata, K. Tsuchihara, S. Mimaki, K. Tsuta, Y. Narita, R. Nishikawa, M. Noguchi, C.C. Harris, A.I. Robles, R. Yamaguchi, S. Imoto, S. Miyano, H. Totsuka, T. Yoshida, and J. Yokota. 2015. Expression and clinical significance of genes frequently mutated in small cell lung cancers defined by whole exome/RNA sequencing. *Carcinogenesis*. 36:616-621.
- Lefebvre, C., T. Bachelot, T. Filleron, M. Pedrero, M. Campone, J.C. Soria, C. Massard, C. Levy, M. Arnedos, M. Lacroix-Triki, J. Garrabey, Y. Boursin, M. Deloger, Y. Fu, F. Commo, V. Scott, L. Lacroix, M.V. Dieci, M. Kamal, V. Dieras, A. Goncalves, J.M. Ferrero, G. Romieu, L. Vanlemmens, M.A. Mouret Reynier, J.C. Thery, F. Le Du, S. Guiu, F. Dalenc, G. Clapissou, H. Bonnefoi, M. Jimenez, C. Le Tourneau, and F. Andre. 2016. Mutational Profile of Metastatic Breast Cancers: A Retrospective Analysis. *PLoS Med*. 13:e1002201.
- Li, Z., N.S. Spoelstra, M.J. Sikora, S.B. Sams, A. Elias, J.K. Richer, A.V. Lee, and S. Oesterreich. 2022. Mutual exclusivity of ESR1 and TP53 mutations in endocrine resistant metastatic breast cancer. *NPJ Breast Cancer*. 8:62.
- Lutchman, M., A.C. Kim, L. Cheng, I.P. Whitehead, S.S. Oh, M. Hanspal, A.A. Boukharov, T. Hanada, and A.H. Chishti. 2002. Dematin interacts with the Ras-guanine nucleotide exchange factor Ras-GRF2 and modulates mitogen-activated protein kinase pathways. *Eur J Biochem*. 269:638-649.
- Mohseni, M., and A.H. Chishti. 2008. The headpiece domain of dematin regulates cell shape, motility, and wound healing by modulating RhoA activation. *Mol Cell Biol*. 28:4712-4718.
- Palaniappan, A., K. Ramar, and S. Ramalingam. 2016. Computational Identification of Novel Stage-Specific Biomarkers in Colorectal Cancer Progression. *PLoS One*. 11:e0156665.
- Remsik, J., Y. Chi, X. Tong, U. Sener, C. Derderian, A. Park, F. Saadeh, T. Bale, and A. Boire. 2022. Leptomeningeal metastatic cells adopt two phenotypic states. *Cancer Rep (Hoboken)*. 5:e1236.
- van Vuren, A., B. van der Zwaag, R. Huisjes, N. Lak, M. Bierings, E. Gerritsen, E. van Beers, M. Bartels, and R. van Wijk. 2019. The Complexity of Genotype-Phenotype Correlations in Hereditary Spherocytosis: A Cohort of 95 Patients: Genotype-Phenotype Correlation in Hereditary Spherocytosis. *Hemasphere*. 3:e276.
- Vasta, J.D., C.R. Corona, J. Wilkinson, C.A. Zimprich, J.R. Hartnett, M.R. Ingold, K. Zimmerman, T. Machleidt, T.A. Kirkland, K.G. Huwiler, R.F. Ohana, M. Slater, P. Otto, M. Cong, C.I. Wells, B.T. Berger, T. Hanke, C. Glas, K. Ding, D.H. Drewry, K.V.M. Huber, T.M. Willson, S. Knapp, S. Muller, P.L. Meisenheimer, F. Fan, K.V. Wood, and M.B. Robers. 2018. Quantitative, Wide-Spectrum Kinase Profiling in Live Cells for Assessing the Effect of Cellular ATP on Target Engagement. *Cell Chem Biol*. 25:206-214 e211.
- Yates, L.R., S. Knappskog, D. Wedge, J.H.R. Farmery, S. Gonzalez, I. Martincorena, L.B. Alexandrov, P. Van Loo, H.K. Haugland, P.K. Lilleng, G. Gundem, M. Gerstung, E. Pappaemmanuil, P. Gazinska, S.G. Bhosle, D. Jones, K. Raine, L. Mudie, C. Latimer, E. Sawyer, C. Desmedt, C. Sotiriou, M.R. Stratton, A.M. Sieuwerts, A.G. Lynch, J.W.

Martens, A.L. Richardson, A. Tutt, P.E. Lonning, and P.J. Campbell. 2017. Genomic Evolution of Breast Cancer Metastasis and Relapse. *Cancer Cell*. 32:169-184 e167.

Ye, Y.P., H.L. Jiao, S.Y. Wang, Z.Y. Xiao, D. Zhang, J.F. Qiu, L.J. Zhang, Y.L. Zhao, T.T. Li, L. Li, W.T. Liao, and Y.Q. Ding. 2018. Hypermethylation of DMTN promotes the metastasis of colorectal cancer cells by regulating the actin cytoskeleton through Rac1 signaling activation. *J Exp Clin Cancer Res*. 37:299.

REVIEWER COMMENTS

Reviewer #1 (Remarks to the Author):

I would like to thank the reviewers for their detailed and satisfactory responses to all the points I raised. I have no further comments.

Reviewer #2 (Remarks to the Author):

Fitzpatrick et al. present a unique dataset with successful characterization of mutations, chromosomal alterations and aberrant expression networks in both plasma, cerebrospinal fluid, primary tumor and other metastases of breast cancer. They successfully grow patient derived organoids from DTs and in PDX. This work has generated a load of valuable data from several clinical and experimental sources.

My comments are again on the clarity of presentation of the results, mainly 1. the order of presentation, 2 that the chapters in which the result section is divided do not always correspond the content and 3. The division of the results is somewhat arbitrary, 4 There is a lot of discussion in the result section that obscures the clarity.

I will give examples:

1. The order of presentation: it will make a much more clear and intriguing reading in my mind if the authors presented all clonal and subclonal heterogeneity in the primary tumor, which clones/subclones are seen there, in what frequencies, then which of these are seen in plasma, which in CSF and DTC, which in the Breast cancer leptomeningeal metastasis (BCLM) and which in other metastases. If all this is given in the context of the primary tumor it will be a lot more clear narrative. As it is I do not see in the result text any place the in depth analysis of the primary tumor. Even if the topic of the paper is the BCLM, it must all come somehow from the primary, so it would be a logical start of the presentation. Perhaps the primary tumor analysis comes in chapter BCLM subclones seed early during tumour evolution, where the authors state:

«To investigate the clonal evolution of BCLM, variant allele frequencies and allele-specific copy number states were used to infer mutational subclones and compose phylogenetic trees (Fig. 3c; Supplementary Fig. 6)»in the primary tumor??? Not stated.

Later:

“However, indicative of a distinct genetic landscape in BCLM, 24.6% of cfDNA mutations across the 11 paired CSF and plasma samples were unique to CSF, and 16.8% were unique to plasma, and 43.4% were shared between both biofluids»The comparison to the primary tumor should have come here.

2. Titles and content of sections under Results

Page 4. Comparative sequencing reveals unique genomic events in BCLM

The above title continues with the finding that the observed aberrations are early event- perhaps this should be the title then with the detailed analysis of the PRIMARY tumor as evidence.

Page 6. Cancer-associated gene aberrations enriched in BCLM

The definition of a “cancer associated gene” escapes me and is too vague for what this paper wants to say. Any somatically aberrant gene is cancer associated as it happened somatically in the cancer. The same crisis occurs when the authors introduce “non-cancer gene alterations”, as gene set enrichment analysis is blind to what genes it will identify, cancer or non-cancer.

Page 8. To assess non-cancer gene alterations acquired in BCLM vs. extracranial metastases, we performed gene set enrichment analysis of frequently altered (by mutation, amplification or deletion) genes in CSF and plasma cfDNA

At the end of chapter BCLM subclones seed early during tumour evolution

Comes a paragraph about including actionable variants found only in CSF cfDNA samples, and shared

with other tissues, alterations with potential therapeutic actionability. This in my mind merits a separate title, as it is of clinical importance and indifferent to the late or early dissemination during tumor evolution.

3. Unnecessary discussion in Results section

page 6: the high rate of CDH1 (E-cadherin) mutations (52%) was an expected finding. Deleterious CDH1 mutations are an early driver event in the majority of invasive lobular breast cancers (ILC), leading to defective adherens junctions^{12,13}. The unexpected finding was deleterious alterations of either CDH1 or another key adherens junction component CTNNA1 (α -catenin) in 55% of the non-lobular BCLM cases (Fig. 2).

Why is this finding second unexpected if found infrequent in non-lobular cases? Perhaps it is novel finding, but not unexpected given the function of catenin?

Page 12: in agreement with published cohorts in breast cancer brain metastasis which show no amplification of ERBB2 but increased ERBB2 expression in brain metastases compared to their matched primary tumours^{48,49}. The increase in HER2 levels in non-amplified (HER2-low) cancers has therapeutic implications given the CNS disease response to HER2-antibody-drug-conjugates (HER2-ADCs)^{50,51}.

Page 12: Again this may reflect the mutual exclusivity of TP53 and ESR1 mutations in metastatic breast cancer⁴⁷

The mutual exclusivity of TP53 and ESR1 mutations reported in reference 47 is related to endocrine resistant metastatic breast cancer, not to all metastatic breast cancer.

47. Li, Z. et al. Mutual exclusivity of ESR1 and TP53 mutations in endocrine resistant metastatic breast cancer. *NPJ Breast Cancer* 8, 62 (2022). <https://doi.org/10.1038/s41523-022-00426-w>

All this inclusion of comments and references works against its goal: instead of convincing the reader, just irritates by obscuring the evidence. If the journal allows for a Result and discussion section together, then re-writing this in the other direction, adding even more discussion may be an option. At the end the authors mention that "intrathecal methotrexate in BCLM, failed to show a significant improvement in survival using this treatment" and that the results presented here indicate other alternatives, but none concrete are mentioned. Perhaps one can highlight the two cases that immune therapy may be an option, or indicate which drugs that penetrate the blood – brain barrier may be of option based on the actionable targets they discover..

Reviewer #3 (Remarks to the Author):

The revised manuscript by Fitzpatrick et al., has addressed some of the concerns raised during the initial reviews. However, there are some specific points that the authors failed to adequately address, which are central to manuscript. First, this paper is describing five novel PDOs derived from DTCs harvested from the CSF. While the authors have injected these 5 PDOs in various sites (orthotopic [MFP], intracardiac), only the KCL320 model was injected directly into the CSF (intracerebroventricular [ICV] inoculation). It would strengthen the paper if the remaining 4 PDOs were injected via the ICV route and the types of leptomeningeal lesions that formed were characterized. Such data would provide the scientific community with a series of PDO models that can form leptomeningeal lesions when injected into mice. This experiment is well within the expertise of the group.

Second, the authors argue that these PDOs represent important models with which to test potential therapeutic strategies. The authors use two inhibitors, crizotinib and taletrectinib. In the initial version of the manuscript, the authors focused on ROS1 has the important receptor tyrosine kinase targeted by these drugs. In the revised manuscript, the authors acknowledge the broader specificity of these inhibitors for multiple tyrosine kinase targets (crizotinib: ROS1, MET, ALK; taletrectinib: ROS, NTRKs). The authors provide a reviewer figure showing the mRNA expression of these various targets in the

different PDOs. It would strengthen the manuscript if the authors could determine which of the proposed targets that are inhibited by these small molecules leads to reduced PDO growth/survival. Are these receptors expressed and activated in the PDOs (immunoblot analyses: total, phospho-tyrosine). Can the authors transiently knockdown these receptors and examine the effect of growth/survival of the PDOs? Do either of these inhibitors impair the growth of PDOs (KCL320) when they are injected into the CSF? These experiments are quite feasible and would add value to the current manuscript. I am still supportive of publication and would be willing to review a revised manuscript with these remaining issues addressed.

Reviewer #4 (Remarks to the Author):

I have had the privilege of reviewing the revised manuscript from Dr. Isacke's lab and carefully examined their meticulously crafted rebuttal letter, which effectively addressed the feedback provided by both myself and my fellow reviewers. The authors' thoughtful and comprehensive responses have resulted in a substantial enhancement of the manuscript.

The incorporation of additional data and the revisions made to the text have undoubtedly led to a significant improvement. The manuscript now conveys its findings in a more coherent manner. The authors have elucidated their methodologies and interpretation of whole exome sequencing (WES) data and copy number analysis, enhancing understanding of these data and the author's interpretations.

Furthermore, the inclusion of patient-derived organoid (PDO) details in the methods section is a positive step forward. However, I would suggest a brief narrative description that highlights the nature of PDOs as 3D Matrigel-embedded cultures, which would aid readers (a minor request). Additionally, while the manuscript contains several early mentions of patient organoids, I recommend that the authors define 'PDO' upon its initial appearance on page 10 (Results, under the 'Development and characterisation of BCLM PDOs' heading).

In summary, the authors' efforts to address reviewer feedback are commendable, and the manuscript now stands as a substantial contribution to the literature on leptomeningeal metastasis. I recommend its publication in Nature Communications.

REVIEWER COMMENTS

Reviewer #1 (Remarks to the Author)

I would like to thank the reviewers for their detailed and satisfactory responses to all the points I raised. I have no further comments.

Response: We thank the reviewer for their positive comments.

Reviewer #2 (Remarks to the Author):

Fitzpatrick et al. present a unique dataset with successful characterization of mutations, chromosomal alterations and aberrant expression networks in both plasma, cerebrospinal fluid, primary tumor and other metastases of breast cancer. They successfully grow patient derived organoids from DTs and in PDX. This work has generated a load of valuable data from several clinical and experimental sources.

My comments are again on the clarity of presentation of the results, mainly 1. the order of presentation, 2 that the chapters in which the result section is divided do not always correspond the content and 3. The division of the results is somewhat arbitrary, 4 There is a lot of discussion in the result section that obscures the clarity.

Response: In the revised manuscript we have heeded these comments and

- (a) provided more information on the evolution of BCLM from the primary tumour - see **new Fig. 1d, Fig. 2, Supplementary Fig. 3; Supplementary Fig. 4**
- (b) re-ordered the manuscript so that the clonal evolution comes before the detailed analysis of the mutations and CNA changes
- (c) created a separate section in the results where the actionable mutations are addressed and a separate section consolidating all of the information (previously scattered through the results section) relating to the aberrations in adherens junction and cytoskeletal components.

Other responses to specific points are provided below

I will give examples:

1. The order of presentation: it will make a much more clear and intriguing reading in my mind if the authors presented all clonal and subclonal heterogeneity in the primary tumor, which clones/subclones are seen there, in what frequencies, then which of these are seen in plasma, which in CSF and DTC, which in the Breast cancer leptomeningeal metastasis (BCLM) and which in other metastases. If all this is given in the context of the primary tumor it will be a lot more clear narrative. As it is I do not see in the result text any place the in depth analysis of the primary tumor. Even if the topic of the paper is the BCLM, it must all come somehow from the primary, so it would be a logical start of the presentation. Perhaps the primary tumor analysis comes in chapter BCLM subclones seed early during tumour evolution, where the authors state:

«To investigate the clonal evolution of BCLM, variant allele frequencies and allele-specific copy number states were used to infer mutational subclones and compose phylogenetic trees (Fig. 3c; Supplementary Fig. 6)»in the primary tumor??? Not stated.

Response: As stated above, we now provide substantial additional information in the clonal evolution figures (**new Fig. 2, new Supplementary Fig. 4 and updated Supplementary Table 6**) and associated text describing the subclone evolution of BCLM from the primary tumour as well as comparing its evolution to that of other metastatic sites.

Later:

“However, indicative of a distinct genetic landscape in BCLM, 24.6% of cfDNA mutations across the 11 paired CSF and plasma samples were unique to CSF, and 16.8% were unique to plasma, and 43.4% were shared between both biofluids»The comparison to the primary tumor should have come here.

Response:

The comparison to the primary tumour is now shown in **new Fig. 1d**, prior to the comparison of CSF to plasma (Fig. 1e)

2. Titles and content of sections under Results

Page 4. Comparative sequencing reveals unique genomic events in BCLM

The above title continues with the finding that the observed aberrations are early event- perhaps this should be the title then with the detailed analysis of the PRIMARY tumor as evidence.

Response: We have now substantially re-ordered and re-written the results section as per the reviewers comments below, and throughout put further emphasis on the comparison to the primary tumour before discussing comparison to other metastatic sites. As a consequence the Results subsection title has now been changed to "Whole exome sequencing reveals unique genomic events in BCLM". Further, we have shown extended clonal composition data in the boxplots and bell plots accompanying phylogenetic trees in **new Figure 2 (3 cases) and new Supplementary Figure 4 (remaining 17 cases)**. These show the subclonal composition of primary tumours with the cancer cell fraction of each clone at time of primary tumour sampling, giving greater detail of the genomic composition of primary tumours in relation to the CSF cfDNA and other sequenced samples.

Page 6. Cancer-associated gene aberrations enriched in BCLM

The definition of a “cancer associated gene” escapes me and is too vague for what this paper wants to say. Any somatically aberrant gene is cancer associated as it happened somatically in the cancer. The same crisis occurs when the authors introduce “non-cancer gene alterations”, as gene set enrichment analysis is blind to what genes it will identify, cancer or non-cancer.

Response: The term "cancer associated gene" has been change to "cancer driver" throughout the manuscript

Page 8. To assess non-cancer gene alterations acquired in BCLM vs. extracranial metastases, we performed gene set enrichment analysis of frequently altered (by mutation, amplification or deletion) genes in CSF and plasma cfDNA

At the end of chapter BCLM subclones seed early during tumour evolution

Comes a paragraph about including actionable variants found only in CSF cfDNA samples, and shared with other tissues, alterations with potential therapeutic actionability. This in my mind merits a separate title, as it is of clinical importance and indifferent to the late or early dissemination during tumor evolution.

Response: We agree with the reviewer and we have created a separate section in the Results on 'actionable variants'

3. Unnecessary discussion in Results section

page 6: the high rate of CDH1 (E-cadherin) mutations (52%) was an expected finding. Deleterious CDH1 mutations are an early driver event in the majority of invasive lobular breast cancers (ILC), leading to defective adherens junctions^{12,13}. The unexpected finding was deleterious alterations of either CDH1 or another key adherens junction component CTNNA1 (α -catenin) in 55% of the non-lobular BCLM cases (Fig. 2).

Why is this finding second unexpected if found infrequent in non-lobular cases? Perhaps it is novel finding, but not unexpected given the function of catenin?

Response: The reason why this finding was unexpected is that mutations in *CDH1* and *CTNNA1* are rare in ductal cancers, whereas here we show that BCLM derived from ductal cancers have acquired such mutations. That said, this section has been rewritten and moved to bring all of the genetic changes in *CDH1* and *CTNNA1* into one place (Results subsection "Enrichment of adherens junction components and cytoskeletal aberrations in BCLM").

Page 12: in agreement with published cohorts in breast cancer brain metastasis which show no amplification of ERBB2 but increased ERBB2 expression in brain metastases compared to their matched primary tumours^{48,49}. The increase in HER2 levels in non-amplified (HER2-low) cancers has therapeutic implications given the CNS disease response to HER2-antibody-drug-conjugates (HER2-ADCs)^{50,51}.

Response: This statement has been moved into the Discussion where new therapeutic strategies are discussed.

Page 12: Again this may reflect the mutual exclusivity of TP53 and ESR1 mutations in metastatic breast cancer⁴⁷

The mutual exclusivity of TP53 and ESR1 mutations reported in reference 47 is related to endocrine resistant metastatic breast cancer, not to all metastatic breast cancer.

47. Li, Z. et al. Mutual exclusivity of ESR1 and TP53 mutations in endocrine resistant metastatic breast cancer. NPJ Breast Cancer 8, 62 (2022). <https://doi.org:10.1038/s41523-022-00426-w>

Response: Apologies, we now clarify this in the text that this reference addresses endocrine therapy-resistant disease.

All this inclusion of comments and references works against its goal: instead of convincing the reader, just irritates by obscuring the evidence. If the journal allows for a Result and discussion section together, then re-writing this in the other direction, adding even more discussion may be an option.

Response: We hope that the reviewer finds that our substantial reorganisation has improved the clarity and readability of the manuscript

At the end the authors mention that "intrathecal methotrexate in BCLM, failed to show a significant improvement in survival using this treatment" and that the results presented here indicate other alternatives, but none concrete are mentioned. Perhaps one can highlight the two cases that immune therapy may be an option, or indicate which drugs that penetrate the blood – brain barrier may be of option based on the actionable targets they discover.

Response: The data on methotrexate treatment of PDOs has been moved into **revised Fig. 6**, and highlights the need for new treatment approaches in BCLM. Accordingly the Discussion has been expanded to include new HER2-ADCs and immunotherapy, based on the findings of enhanced HER2 IHC scores and/or high TMB in certain cases within our cohort.

Reviewer #3 (Remarks to the Author):

The revised manuscript by Fitzpatrick et al., has addressed some of the concerns raised during the initial reviews. However, there are some specific points that the authors failed to adequately address, which are central to manuscript. First, this paper is describing five novel PDOs derived from DTCs harvested from the CSF. While the authors have injected these 5 PDOs in various sites (orthotopic [MFP], intracardiac), only the KCL320 model was injected directly into the CSF (intracerebroventricular [ICV] inoculation). It would strengthen the paper if the remaining 4 PDOs were injected via the ICV route and the types of leptomeningeal lesions that formed were characterized. Such data would provide the scientific community with a series of PDO models that can form leptomeningeal lesions when injected into mice. This experiment is well within the expertise of the group.

Response: Such an experiment is within our expertise but it is an expensive and time-consuming experiment that will take longer than the 4 weeks deadline Nat Comms have requested for a revised submission. More importantly, we believe it would add little if anything to the main conclusions of this part of the work i.e. that it is possible to generate PDOs from the rare CSF DTC, that these PDOs can be expanded in culture and can grow as PDXs in mice and in most cases exhibit growth in the leptomeninges when injected by intracardiac route. When injected into a restricted niche such as the CSF, the PDOs (as we showed for KCL320) will likely grow within that site given that it is the site from which they were derived.

Second, the authors argue that these PDOs represent important models with which to test potential therapeutic strategies. The authors use two inhibitors, crizotinib and taletrectinib. In the initial version of the manuscript, the authors focused on ROS1 has the important receptor tyrosine kinase targeted by these drugs. In the revised manuscript, the authors acknowledge the broader specificity of these inhibitors for multiple tyrosine kinase targets (crizotinib: ROS1, MET, ALK; taletrectinib: ROS, NTRKs). The authors provide a reviewer figure showing the mRNA expression of these various targets in the different PDOs. It would strengthen the manuscript if the authors could determine which of the proposed targets that are inhibited by these small molecules leads to reduced PDO growth/survival. Are these receptors expressed and activated in the PDOs (immunoblot analyses: total, phospho-tyrosine). Can the authors transiently knockdown these receptors and examine the effect of growth/survival of the PDOs? Do either of these inhibitors impair the growth of PDOs (KCL320) when they are injected into the CSF? These experiments are quite feasible and would add value to the current manuscript. I am still supportive of publication and would be willing to review a revised manuscript with these remaining issues addressed.

Response: We state in our manuscript that the development of PDOs derived from tumour cells within the CSF provides a valuable resource for testing of therapeutic reagents and used the examples of methotrexate treatment (given that methotrexate is the current standard of care treatment for BCLM patients) and crizotinib/taletrectinib treatment (given the reports from other laboratories that these inhibitors are synthetic lethal with loss of E-cadherin). The reviewer asks an interesting question about the specificity of crizotinib/taletrectinib but it is beyond the scope of this manuscript to explore their mechanism of action. Given the reviewer's comments, and after discussion with the Editor, we feel that the best course of action is to

remove the crizotinib/taletrectinib data and show only the methotrexate data, which has now moved to **revised Fig. 6**.

Reviewer #4 (Remarks to the Author):

I have had the privilege of reviewing the revised manuscript from Dr. Isacke's lab and carefully examined their meticulously crafted rebuttal letter, which effectively addressed the feedback provided by both myself and my fellow reviewers. The authors' thoughtful and comprehensive responses have resulted in a substantial enhancement of the manuscript.

The incorporation of additional data and the revisions made to the text have undoubtedly led to a significant improvement. The manuscript now conveys its findings in a more coherent manner. The authors have elucidated their methodologies and interpretation of whole exome sequencing (WES) data and copy number analysis, enhancing understanding of these data and the author's interpretations.

Furthermore, the inclusion of patient-derived organoid (PDO) details in the methods section is a positive step forward. However, I would suggest a brief narrative description that highlights the nature of PDOs as 3D Matrigel-embedded cultures, which would aid readers (a minor request). Additionally, while the manuscript contains several early mentions of patient organoids, I recommend that the authors define 'PDO' upon its initial appearance on page 10 (Results, under the 'Development and characterisation of BCLM PDOs' heading).

In summary, the authors' efforts to address reviewer feedback are commendable, and the manuscript now stands as a substantial contribution to the literature on leptomeningeal metastasis. I recommend its publication in Nature Communications.

Response: We thank the reviewer for their positive comments. With regards the specific requests we have (a) defined PDO in the heading on page 12. We are only allowed to define an abbreviation once, and (b) expanded this section of the results to both encourage the reader to see the Methods section where we have provided full details of the media composition and PDO handling - and to clarify that the PDOs were grown and passaged as Matrigel-embedded cultures.

REVIEWERS' COMMENTS

Reviewer #2 (Remarks to the Author):

The article is rewritten to considerably higher clarity. The inclusion of the tumor clonality data and figure 2 adds a lot.

The compendium of generated data will advance the field of metastatic breast cancer further, especially with these rare, hard to reach and clinically very significant lesions. I have no further comments. Hope and look forward to read the article in the journal.

Reviewer #3 (Remarks to the Author):

The authors have submitted a second revision for the manuscript entitled "Genomic profiling and pre-clinical modelling of breast cancer leptomeningeal metastasis reveals acquisition of a lobular-like phenotype". The authors have been responsive to issues raised by the other reviewers. With respect to the additional comments I have raised, the assessment of the additional patient-derived organoids was not performed due to the length of time this would take, precluding the submission of a revised manuscript in the required timeframe.

The second point I raised was the targets for the two inhibitors that were used in the study. The authors argue that delineating the precise targets of these inhibitors was beyond the scope and, in consultation with the editor, have decided to remove the crizotinib and taletrectinib data from the manuscript, and focus only on the methotrexate data, which is fine. Given the overall responses to the additional 3 reviewers, which have improved the clarity of the manuscript, I support publication in Nature Communications.

REVIEWER COMMENTS

Reviewer #2 (Remarks to the Author):

The article is rewritten to considerably higher clarity. The inclusion of the tumor clonality data and figure 2 adds a lot.

The compendium of generated data will advance the field of metastatic breast cancer further, especially with these rare, hard to reach and clinically very significant lesions. I have no further comments. Hope and look forward to read the article in the journal.

Response: We thank the reviewer for their positive comments.

Reviewer #3 (Remarks to the Author):

The authors have submitted a second revision for the manuscript entitled "Genomic profiling and pre-clinical modelling of breast cancer leptomeningeal metastasis reveals acquisition of a lobular-like phenotype". The authors have been responsive to issues raised by the other reviewers. With respect to the additional comments I have raised, the assessment of the additional patient-derived organoids was not performed due to the length of time this would take, precluding the submission of a revised manuscript in the required timeframe.

The second point I raised was the targets for the two inhibitors that were used in the study.

The authors argue that delineating the precise targets of these inhibitors was beyond the scope and, in consultation with the editor, have decided to remove the crizotinib and taletrectinib data from the manuscript, and focus only on the methotrexate data, which is fine. Given the overall responses to the additional 3 reviewers, which have improved the clarity of the manuscript, I support publication in Nature Communications.

Response: We thank the reviewer for their positive comments.